# Exploring Social Posterior Collapse in Variational Autoencoder for Interaction Modeling

**Chen Tang**
Department of Mechanical Engineering
University of California Berkeley
Berkeley, CA
chen_tang@berkeley.edu

**Wei Zhan**
Department of Mechanical Engineering
University of California Berkeley
Berkeley, CA
wzhan@berkeley.edu

**Masayoshi Tomizuka**
Department of Mechanical Engineering
University of California Berkeley
Berkeley, CA
tomizuka@berkeley.edu

## Abstract

Multi-agent behavior modeling and trajectory forecasting are crucial for the safe navigation of autonomous agents in interactive scenarios. Variational Autoencoder (VAE) has been widely applied in multi-agent interaction modeling to generate diverse behavior and learn a low-dimensional representation for interacting systems. However, existing literature did not formally discuss if a VAE-based model can properly encode interaction into its latent space. In this work, we argue that one of the typical formulations of VAEs in multi-agent modeling suffers from an issue we refer to as social posterior collapse, i.e., the model is prone to ignoring historical social context when predicting the future trajectory of an agent. It could cause significant prediction errors and poor generalization performance. We analyze the reason behind this under-explored phenomenon and propose several measures to tackle it. Afterward, we implement the proposed framework and experiment on real-world datasets for multi-agent trajectory prediction. In particular, we propose a novel sparse graph attention message-passing (sparse-GAMP) layer, which helps us detect social posterior collapse in our experiments. In the experiments, we verify that social posterior collapse indeed occurs. Also, the proposed measures are effective in alleviating the issue. As a result, the model attains better generalization performance when historical social context is informative for prediction.

## 1 Introduction

Modeling the behavior of intelligent agents is an essential subject for autonomous systems. Safe operations of autonomous agents require accurate prediction of other agents' future motions. In particular, social interaction among agents should be modeled, so that downstream planning and decision-making modules can safely navigate the robots through interactive scenarios. Generative latent variable models are popular modeling options for their ability to generate diverse and naturalistic behavior [1, 2, 3, 4]. In this work, we focus on one category of generative models, Variational Autoencoder (VAE) [5], which has been widely used in multi-agent behavior modeling and trajectory prediction [1, 4, 6, 7, 8, 9]. It is desirable as it learns a low-dimensional representation of the original high-dimensional data.

35th Conference on Neural Information Processing Systems (NeurIPS 2021)

However, VAEs do not necessarily learn a good representation of the data [10]. For instance, prior works in sequence modeling have found out that the model tends to ignore the latent variables if the decoder is overly powerful [11, 12, 13] (e.g., an autoregressive decoder). It leads us to wonder whether a VAE-based model can always learn a good representation for a multi-agent interacting system. It is a rather general question as researchers may look for different properties of the latent space, for instance, interpretability [14, 15] and multi-modality [4, 6]. In this work, we focus on a fundamental aspect of this general question: Does the latent space always properly model interaction? Formally, given a latent variable model of an interacting system, where a latent variable governs each agent's behavior, we wonder if the VAE learns to encode social context into the latent variables.

It raises such concern since VAE handles two distinct tasks in training and testing. At the training stage, it learns to reconstruct a datum instead of generating one. For multi-agent interaction modeling, the sample for reconstruction is a set of trajectories of all the agents. Ideally, the model should learn to model the interaction among agents and jointly reconstruct the trajectories. However, there is no mechanism to prevent the model from separately reconstructing the trajectories. In fact, it could even be more efficient at the early stage of training when the model has not learned an informative embedding of social context. We then end up with a model that models each agent's behavior separately without social context, which might suffer from over-estimated variance and large prediction error. More importantly, since the joint behavior of the agents is a consequence of their interactions, ignoring the causes may lead to poor generalization ability [15, 16]. We find that a typical formulation of VAE for multi-agent interaction is indeed prone to ignoring historical social context (i.e., interactions over the observed time horizon). We refer to this phenomenon as social posterior collapse. The issue has never been discussed in the literature. Considering those potential defects, we think it is necessary to study such a crucial and fundamental issue.

In this work, we first abstract the VAE formulation from a wide range of existing literature [1, 4, 6, 8]. Then we analyze social posterior collapse under this formulation and propose several measures to alleviate the issue. Afterward, we study the issue under real-world settings with a realization of the abstract formulation we design. In particular, we propose a novel sparse graph attention message-passing (sparse-GAMP) layer and incorporate it into the model, which helps us detect and analyze social posterior collapse. Our experiments show that social posterior collapse occurs in real-world prediction tasks and that the proposed measures can effectively alleviate the issue. We also evaluate how social posterior collapse affects the model performance on these tasks. The results suggest that the model without social posterior collapse can attain better generalization performance if historical social context is informative for prediction.

## 2 Social Posterior Collapse in Variational Autoencoder Interaction Models

### 2.1 A Latent Variable Model for Interaction Modeling

Given an interacting system with $n$ agents, an interaction-aware trajectory prediction model takes all the agents' observed historical trajectories, denoted by $\{\mathbf{x}_i\}_{i=1}^n$, as input and predicts the future trajectories of all the agents or a subset of enquired agents. We denote the collection of future trajectories by $\{\mathbf{y}_i\}_{i=1}^n$. In this work, we focus on the latent variable model illustrated in Fig. 1, which is abstracted from existing literature on VAEs for multi-agent behavior modeling [1, 4, 6, 8]. We model interaction by introducing a set of variables $\{\mathbf{T}_i\}_{i=1}^n$, which aggregates each agent's state and its observation of other agents. Formally, each $\mathbf{T}_i$ is modeled as a deterministic function of $\{\mathbf{x}_i\}_{i=1}^n$, i.e., $\mathbf{T}_i = f_i(\{\mathbf{x}_i\}_{i=1}^n)$. Afterward, the agents make decisions based on the aggregated information over the predicted horizon. Latent variables $\{\mathbf{z}_i\}_{i=1}^n$ are introduced to model the inherent uncertainty in each agent's behavior. It should be noticed that interaction over the predicted horizon is not modeled explicitly in this formulation. Although it can be achieved by exchanging information between agents recurrently (e.g., social pooling in [17]), it is a common practice to avoid explicit modeling of future interaction in consideration of computational cost and memory [3, 8, 18].

In this work, we train the model as a VAE, where an encoder $q(\mathbf{z}|\mathbf{x},\mathbf{y})$ is introduced to approximate the posterior distribution $p(\mathbf{z}|\mathbf{x},\mathbf{y})$ for efficient sampling at the training stage[1]. The performance of variational inference is optimized if the KL-divergence between the posteriors, i.e. $D_{KL}[q(\mathbf{z}|\mathbf{x},\mathbf{y})\|p(\mathbf{z}|\mathbf{x},\mathbf{y})]$, is small [10]. To derive a better approximation, we can incorporate

---

[1]The vectors $\mathbf{x}$, $\mathbf{y}$ and $\mathbf{z}$ collect the corresponding variables for all $n$ agents.

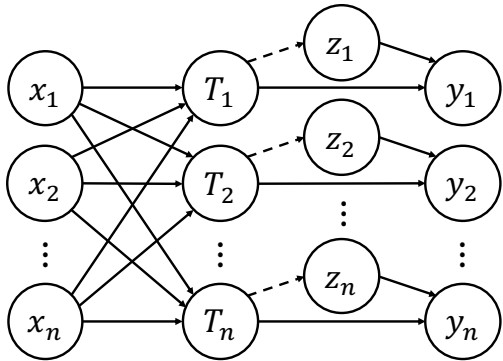

Figure 1: A latent variable model for interaction modeling. The dash edges from $T_i$ to $z_i$ apply only in the Conditional Variational Autoencoder (CVAE) setting.

inductive bias based on the characteristics of the true posterior into the encoder function. We introduce the following simple proposition that guides our model design. Its proof can be found in Appx. A.

**Proposition 1.** *For any $i = 1, 2, ..., n$ and $j = 1, 2, ..., n$, 1) If $j \neq i$, $\mathbf{z}_i$ and $\mathbf{y}_j$ are conditionally independent given $\mathbf{x}$ and $\mathbf{y}_i$; 2) If $j \neq i$, $\mathbf{z}_i$ and $\mathbf{z}_j$ are conditionally independent given $\mathbf{x}$; 3) $\mathbf{z}_i$ and $\mathbf{x}_j$ are conditionally independent given $\mathbf{T}_i$.*

Following the proposition, we decompose the posterior distribution as $\prod_{i=1}^{n} p(\mathbf{z}_i | \mathbf{T}_i, \mathbf{y}_i)$. The decomposition suggests two insights. First, the encoder does not need to aggregate future context information when inferring the posterior distribution of $\mathbf{z}_i$ for each agent. Second, the historical context variable $\mathbf{T}_i$ is all we need for the historical information of the agent $i$. The encoder and decoder can share the same function to encode historical information.

## 2.2 Social Posterior Collapse

We then design a VAE model reflecting the characteristics of the true posterior discovered in Sec. 2.1. To simplify the problem, we further make the assumption of homogeneous agents. The model has three basic building blocks: 1) A function modeling the historical context, i.e., $\mathbf{T}_i = f_\theta (\mathbf{x}_i, \mathbf{x})$; 2) A function decoding the distribution of the future trajectory $\mathbf{y}_i$ given $\mathbf{T}_i$ and $\mathbf{z}_i$, i.e., $p_\phi (\mathbf{y}_i | \mathbf{T}_i, \mathbf{z}_i)$; 3) A function approximating the posterior of $\mathbf{z}_i$ conditioned on $\mathbf{T}_i$ and $\mathbf{y}_i$, i.e., $q_\psi (\mathbf{z}_i | \mathbf{T}_i, \mathbf{y}_i)$. They build up the encoder and decoder of the VAE model as follows:

$$q_{\theta,\psi} (\mathbf{z}|\mathbf{x}, \mathbf{y}) = \prod_{i=1}^{n} q_\psi (\mathbf{z}_i | f_\theta(\mathbf{x}_i, \mathbf{x}), \mathbf{y}_i), \qquad p_{\theta,\phi} (\mathbf{y}|\mathbf{x}, \mathbf{z}) = \prod_{i=1}^{n} p_\phi (\mathbf{y}_i | f_\theta(\mathbf{x}_i, \mathbf{x}), \mathbf{z}_i). \quad (1)$$

We consider a continuous latent space and model $q_\psi$ and $p_\phi$ as diagonal Gaussian distributions. The model is trained by maximizing the evidence lower bound (ELBO):

$$\max_{\theta,\psi,\phi} \mathbb{E}_{\mathbf{x},\mathbf{y} \sim D} \left[ \mathbb{E}_{\mathbf{z} \sim q_{\theta,\psi}(\mathbf{z}|\mathbf{x},\mathbf{y})} \left[ \log p_{\theta,\phi} (\mathbf{y}|\mathbf{x}, \mathbf{z}) \right] - \beta D_{KL} \left[ q_{\theta,\psi} (\mathbf{z}|\mathbf{x}, \mathbf{y}) \| p(\mathbf{z}) \right] \right]. \quad (2)$$

However, we find a critical issue of this naive formulation during experiments, which is the social posterior collapse phenomenon mentioned before. With the help of the sparse graph attention mechanism introduced in Sec. 3, we find that the model is prone to ignoring historical social context when reconstructing the future trajectory of one agent. Equivalently, the generative model collapses to the one with all the variables of other agents marginalized out.

We think the reason behind it is similar to, but different from, the well-known phenomenon in VAE training—posterior collapse [10]. Due to the KL regularization term in ELBO, the variational posterior distribution is likely to collapse towards the prior. It is particularly likely to occur at the early stage of training when the latent space is still uninformative [19]. In our case, the posterior of $\mathbf{z}_i$ collapses into $q_{\theta,\psi}(\mathbf{z}_i | \mathbf{x}_i, \mathbf{y}_i)$. It does minimize the KL-divergence: if both $q_{\theta,\psi} (\mathbf{z}_i | \mathbf{x}_i, \mathbf{y}_i)$ and $q_{\theta,\psi}(\mathbf{z}_i | \mathbf{x}, \mathbf{y}_i)$ exactly approximate the true posteriors, then conditioning on more context information increases the expected value of KL regularization:

$$\mathbb{E}_D \left[ D_{KL} \left[ p (\mathbf{z}_i | \mathbf{x}, \mathbf{y}_i) \| p(\mathbf{z}_i) \right] \right] = I(\mathbf{z}_i; \mathbf{x}, \mathbf{y}_i) \geqslant I(\mathbf{z}_i; \mathbf{x}_i, \mathbf{y}_i) = \mathbb{E}_D \left[ D_{KL} \left[ p (\mathbf{z}_i | \mathbf{x}_i, \mathbf{y}_i) \| p(\mathbf{z}_i) \right] \right].$$

However, we argue that an additional factor contributes to the social posterior collapse problem, which makes it a unique phenomenon for interaction modeling. At the training stage, the goal is to reconstruct the future trajectories $\mathbf{y}$. The trajectory itself contains all the information needed for reconstruction. The history of the agent $i$ provides complementary information such as the current state and static environmental information. There is no explicit regulation in the current framework to prevent the model from extracting information solely from $\mathbf{x}_i$ and $\mathbf{y}_i$. In fact, it is a more efficient coding scheme when the model has not learned an informative representation of interaction. Techniques such as KL annealing [13, 19] rely on various scheduling schemes of $\beta$ to prevent KL vanishing, which has been shown effective in mitigating the posterior collapse problem. However, reducing $\beta$ could merely privilege the model to gather more information from $\mathbf{y}_i$, which is consistent with the reconstruction objective. Therefore, we need to explore alternative solutions to tackle the social posterior collapse problem deliberately.

We start with changing the model into a conditional generative one [20, 21]. Additional edges $\{\mathbf{T}_i \rightarrow \mathbf{z}_i\}_{i=1}^n$ are added into the original graph, which are annotated with dash lines in Fig. 1. We follow the practice in [20] and formulate the model as a Conditional Variational Autoencoder (CVAE). It is straightforward to verify that the conclusion of Proposition 2.1 still applies. We still model the encoder and decoder as in Eqn. (1). The CVAE framework introduces an additional module —a function approximating the conditional prior $p_\eta(\mathbf{z}_i|\mathbf{T}_i)$, which becomes $p_{\theta,\eta}(\mathbf{z}_i|\mathbf{x})$ after incorporating $f_\theta(\mathbf{x})$. The objective then becomes:

$$\mathcal{L}(\theta, \psi, \phi, \eta) = \mathbb{E}_{\mathbf{x},\mathbf{y}\sim D}\left[\mathbb{E}_{\mathbf{z}\sim q_{\theta,\psi}(\mathbf{z}|\mathbf{x},\mathbf{y})}\left[\log p_{\theta,\phi}(\mathbf{y}|\mathbf{x},\mathbf{z})\right] - \beta D_{KL}\left[q_{\theta,\psi}(\mathbf{z}|\mathbf{x},\mathbf{y})\|p_{\theta,\eta}(\mathbf{z}|\mathbf{x})\right]\right].$$

Compared to Eqn. (2), we no longer penalize the encoder for aggregating context information but only restrict the information encoded from future trajectories. Therefore, the encoder does not need to ignore context information to fulfill the information bottleneck. However, the model still lacks a mechanism to encourage context information encoding deliberately. To achieve the goal, we propose to incorporate an auxiliary prediction task into the training scheme. Concretely, we introduce another module $\mathbf{y}_i = g_\zeta(\mathbf{T}_i, \mathbf{z}_i)$. Composing $g_\zeta$ with $f_\theta$ gives us another decoder $\mathbf{y} = h_{\theta,\zeta}(\mathbf{x}, \mathbf{z})$. The difference is that the latent variables are always sampled from $p_\eta(\mathbf{z}_i|\mathbf{T}_i)$. The auxiliary task is training this trajectory decoder which shares the same context encoder and conditional prior with the CVAE. Because the auxiliary model does not have access to the ground-truth future trajectory, it needs to utilize context information for accurate prediction. Consequently, it encourages the model to encode context information into $\mathbf{T}$ and $\mathbf{z}$. We use mean squared error (MSE) loss as the objective function. The overall objective minimizes a weighted sum of the two objective functions.

The training scheme looks similar to the one in [20], where they trained a Gaussian stochastic neural network (GSNN) together with the CVAE model. However, ours is different from theirs in some major aspects. The GSNN model shares the same decoder with the CVAE model. The primary motivation of incorporating another learning task is to optimize the generation procedure during training directly. In contrast, our auxiliary prediction model has a separate trajectory decoder. It is because we do not want the auxiliary task to interfere with the decoding procedure of the CVAE model. We find that sharing the decoder leads to less diversity in trajectory generation, which is not desirable.

## 3 Sparse Graph Attention Message-Passing Layer

Before jumping to introducing the specific model we develop for real-world prediction tasks, we would like to present a novel sparse graph attention message-passing (sparse-GAMP) layer, which helps us detect and analyze the social posterior collapse phenomenon.

### 3.1 Sparse Graph Attention Mechanism

Our sparse-GAMP layer incorporates $\alpha$-entmax [22] as the graph attention mechanism. $\alpha$-entmax is a sparse transformation that unifies softmax and sparsemax [23]. Sparse activation functions have drawn growing attention recently because they can induce sparse and interpretable outputs. In the context of VAEs, they have been used to sparsify discrete latent space for efficient marginalization [24] and tractable multimodal sampling [25]. In our case, we mainly use $\alpha$-entmax to induce a sparse and interpretable attention map within the encoder for diagnosing social posterior collapse.

We are particularly interested in the 1.5-entmax variant, which is smooth and can be exactly computed. It is also easy to implement on GPUs using existing libraries (e.g., PyTorch [26]). Concretely, the 1.5-entmax function maps a $d$-dimensional input $\mathbf{s} \in \mathbb{R}^d$ into $\mathbf{p} \in \Delta^d = \left\{ \mathbb{R}^d : \mathbf{p} \geqslant 0, \|\mathbf{p}\|_1 = 1 \right\}$ as $\mathbf{p} = [\mathbf{s}/2 - \tau \mathbf{1}]_+^2$, where $\tau$ is a unique threshold value computed using $\mathbf{s}$. Upon the theoretical results in [22], we derive an insightful proposition which makes 1.5-entmax a merited option in our framework. The proof of the proposition can be found in Appx. B.

**Proposition 2.** *Let $s_{[d]} \leqslant \cdots \leqslant s_{[1]}$ denote the sorted coordinates of $\mathbf{s}$. Define the top-$\rho$ mean, unnormalized variance, and induced threshold for $\rho \in \{1, ..., d\}$ as*

$$M_{\mathbf{s}}(\rho) = \frac{1}{\rho} \sum_{j=1}^{\rho} s_{[j]}, \ S_{\mathbf{s}}(\rho) = \sum_{j=1}^{\rho} \left( s_{[j]} - M_{\mathbf{s}}(\rho) \right)^2, \ \tau_{\mathbf{s}}(\rho) = \begin{cases} M_{\mathbf{s}}(\rho) - \sqrt{\frac{1 - S_{\mathbf{s}}(\rho)}{\rho}}, & S_{\mathbf{s}}(\rho) \leqslant 1, \\ +\infty, & otherwise. \end{cases}$$

*Let $\mathbf{s}' \in \mathbb{R}^{d+1}$ satisfy $s_i' = s_i$ for $i \leqslant d$, and define $\mathbf{p} = 1.5\text{-entmax}(\mathbf{s})$ and $\mathbf{p}' = 1.5\text{-entmax}(\mathbf{s}')$.*

*Then we have: 1) If $\frac{s_{d+1}'}{2} \leqslant \frac{s_{[d]}}{2} - 1$, then $p_i' = p_i$ for $i = 1, ..., d$ and $p_{d+1}' = 0$; 2) If $p_i > 0$ for $i = 1, .., d$, then $p_i' = p_i$ for $i = 1, 2, ..., d$ and $p_{d+1}' = 0$ iff $s_{d+1}' \leqslant 2\tau_{\mathbf{s}/2}(d)$.*

The first statement of the proposition provides a sufficient condition for augmenting an input vector without affecting its original attention values. It is useful when applying 1.5-entmax to graph attention. Unlike typical neural network models, graph neural networks (GNN) operate on graphs whose sizes vary over different samples. Meanwhile, nodes within the same graph may have different numbers of incoming edges. Therefore, the 1.5-entmax function has inputs of varying dimensions even within the same batch of training samples, making it inefficient to compute using available primitives. The proposition suggests a simple solution to this problem. Given $\{\mathbf{s}_j\}_{j=1}^m$ with $\mathbf{s}_j \in \mathbb{R}^{d_j}$, we can compute a dummy value as $\min_{j \in \{1, ..., m\}, i \in \{1, ..., d_j\}} s_{j,i} - 2$. We can augment the input vectors with this dummy value to transform them into a matrix in $\mathbb{R}^{m \times d^*}$, where $d^*$ is the largest value of $d_j$. The dummy elements will not affect the attention computation, and existing primitives based on matrix computation can be directly used.

The second statement implies that the activated coordinates determine a unique threshold value for augmented coordinates. This property is beneficial when modeling interactions with a large number of agents (e.g., dense traffic scenes). It ensures that the effects of interacting agents will not be diluted by the irrelevant agents, which could potentially improve the robustness of the model [27]. In this work, we mainly utilize the interpretability of the sparse graph attention, but we will investigate its application in generalization in future work.

## 3.2 Sparse-GAMP

To obtain the sparse-GAMP layer, we combine the sparse graph attention with a message-passing GNN [28, 29]. Given a directed graph $\mathcal{G} = (\mathcal{V}, \mathcal{E})$ with vertices $v \in \mathcal{V}$ and edges $e = (v, v') \in \mathcal{E}$. we define the sparse-GAMP layer as a composition of a node-to-edge message-passing step $v \to e$ and an edge-to-node message-passing step $e \to v$:

$$v \to e : \ \mathbf{h}_{(i,j)} = f_e \left( \left[ \mathbf{h}_i, \mathbf{h}_j, \mathbf{u}_{(i,j)} \right] \right), \tag{3}$$

$$e \to v : \quad \hat{\mathbf{h}}_j = \sum_{i \in \mathcal{N}_j} w_{(i,j)} \mathbf{h}_{(i,j)}, \ \text{where} \ \mathbf{w}_j = \mathcal{G}\text{-entmax} \left( \left\{ \mathbf{h}_{(i,j)} \right\}_{i \in \mathcal{N}_j} \right),$$

In our prediction model, we will use this sparse-GAMP layer to model the function for historical social context encoding, i.e., $\mathbf{T}_i = f_\theta(\mathbf{x}_i, \mathbf{x})$.

## 4 Social-CVAE

In this section, we design a realization of the abstract framework studied in Sec. 2 for real-world trajectory prediction tasks. We choose to design the model with GNNs, which enable a flexible graph representation of data. Several GNN-based approaches achieved the state-of-the-art performance on trajectory prediction task [30, 31, 32]. The resulting model is depicted in Fig. 2, which we refer to as social-CVAE. During training, we first encode the historical and future trajectories of all the agents using a Gated Recurrent Unit (GRU) [33] network. For vehicle trajectory prediction tasks,

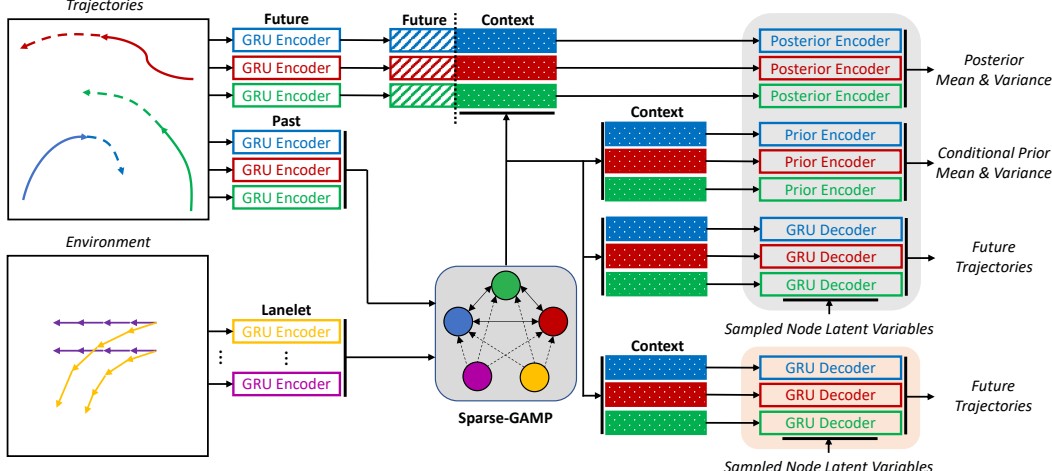

Figure 2: Social-CVAE architecture. We adopt a graph representation and use sparse-GAMP to encode context information. The modules within the orange box are the auxiliary decoders.

we incorporate the map information in a manner similar to [34]. Different from [34] where road boundaries are modeled as nodes in the scene graph, we adopt the representation in [35] where the map is denoted as a graph consisting of lanelet nodes, i.e., drivable road segments. For each lanelet, it is represented by its left and right boundaries composed by two sequences of points. We use another GRU network to encode them. Although not utilized in this work, the lanelet representation allows us to combine routing information in a natural manner as in [32]. We will investigate it in future work.

To encode the historical context information, we construct a graph using the embeddings of historical trajectories and lanelets if available. Each pair of agent nodes has a bidirectional edge connecting each other. For each lanelet node, we add edges connecting it to all the agent nodes. If the map information is not available, we add self-edges for all the agents to enable direct self-encoding channels. One sparse-GAMP layer is applied to encode historical context for all the agents. If social posterior collapse occurs, the agent-to-agent edges, except for self-edges, are more likely to receive zero attention weights thanks to the sparse attention. Therefore, we can use the percentage of unattended agent-to-agent edges as a metric to detect social posterior collapse. It is a more objective metric than looking into the quantitative magnitude of attention weights. We can assert that an agent node does not contribute to the output if its attention is strictly zero. However, we need to be more careful when comparing the importance of two agents based on non-zero attention weights [36, 37]. For the same reason, we do not consider techniques such as multiple message-passing layers [34, 31, 32] or separating self-edges from aggregation [8]. Even though they can potentially boost the performance, it might become inconclusive to analyze the model based on the attention map. After computing the historical context, we use a MLP to model the conditional Gaussian prior $p_\eta(\mathbf{z}_i|\mathbf{T}_i)$. To model the variational posterior, we concatenate the future trajectory embedding with each agent's historical context and use another MLP to output the posterior mean and variance. For the decoder, we use another GRU to decode the future trajectory for each agent separately. The auxiliary decoder shares the same structure but always samples the latent variables from the conditional prior regardless of training or testing.

## 5    Experiments

In this section, we report the experimental results on two trajectory prediction tasks. The main purpose is not achieving state-of-the-art performance but to study the social posterior collapse problem. We compare social-CVAE with two variants: 1) A model without the auxiliary task; 2) A model without the auxiliary task or the conditional prior. They correspond to the vanilla CVAE and VAE formulations discussed in Sec.2. We are curious about whether the proposed measures can alleviate social posterior collapse. We will briefly introduce the experiment settings and present the main results. Please refer to Appx. C-F for more details, visualization, and additional experiments.

## 5.1 Vehicle Trajectory Prediction

The first task is the vehicle trajectory prediction problem, where we are asked to predict the future trajectory of one target vehicle given the historical trajectories of itself and other surrounding vehicles. We train the models on two datasets, INTERACTION Dataset [38] and Argoverse Motion Forecasting dataset [39]. To evaluate the prediction performance, we use the standard minimum Average Displacement Error ($min$ADE) and minimum Final Displacement Error ($min$FDE) over $K$ sampled trajectories as metrics. And we follow [39] to define $min$ADE as ADE of the trajectory with minimum FDE. Additionally, we define a unique metric, Agent Ratio (AR), to study social posterior collapse:

$$\text{AR} = \frac{\sum_{i=1, i \neq j}^{n} \mathbf{1}(\omega_{(i,j)} \neq 0)}{n - 1},$$

where the agent $j$ refers to the predicted target vehicle and $w_{(i,j)}$ is the attention weight assigned to the edge from the agent $i$ to the target vehicle. It equals to the percentage of surrounding vehicles which receive non-zero attention. A value close to zero implies that the model ignores the majority of the surrounding vehicles, which is a sign of social posterior collapse.

**INTERACTION Dataset.** The INTERACTION dataset provides three categories of driving scenarios: intersection (IS), roundabout (RA), and highway merging (HM). For each experiment, we ran five trials to account for the randomness in initialization. We then evaluated them on the validation sets and computed the mean and standard deviation of the evaluated metrics over all the trials. The best models were selected for testing on the regular (R) and generalization (G) tracks of the INTERPRET Challenge [2]. The results are summarized in Table 1. The CVAE variants have AR values similar to the VAE variants on IS and HM. It is consistent with our argument that merely changing the model to a conditional one is insufficient.

In contrast, our social-CVAE model consistently attains high AR values. However, the CVAE variant achieves similar prediction performance as ours on the validation sets of IS and RA, even if the CVAE variant suffers from the social posterior collapse problem. It is because the driving behavior within intersections and roundabouts depends highly on locations. When the map is less informative (e.g., validation set in HM) or a novel scenario is encountered (e.g., generalization track), our social-CVAE model, which does not have social posterior collapse outperforms the other variants. Notably, we compare it with another instance that attains a AR value close to zero even with the auxiliary task. Their test performances, especially on the generalization track, are pretty different. It shows that it is mainly the historical social context that improves the prediction performance. Even if the auxiliary task also contributes, the improvement is not significant, especially in novel scenes, unless it prevents social posterior collapse.

**Argoverse Dataset.** Similar to the INTERACTION dataset, we trained five models with random initialization for each case and evaluated them on the validation sets. The best social-CVAE instance was selected for submission to the Argoverse Motion Forecasting Challenge. Although achieving state-of-the-art performance is not our objective, we still report the testing result in Appx. F and compare it with the other models on the leaderboard in order to provide the audience a complete picture of the model. Here, we mainly focus on the validation results in Table 2. The conclusion is consistent. The difference is that the social-CVAE model outperforms the others even on the validation set. It is because the validation set of the Argoverse dataset is collected in different areas of the cities, which is more analogous to the generalization track of the INTERPRET Challenge.

## 5.2 Pedestrian Trajectory Prediction

The second task is the pedestrian trajectory prediction problem, where we are asked to forecast the future trajectories of all the pedestrians in each scenario. We trained the models on the well-established ETH/UCY dataset, which combines two datasets, ETH [40] and UCY [41]. Following prior works [2, 42, 4, 7, 9], we adopt the leave-one-out evaluation protocol and do not use any environmental information. All the prediction metrics are computed with $K = 20$ samples. Similar to the vehicle case, we ran five trials for each experiment. The results are summarized in Table 3, where we report the testing results for each group and their average over all the groups. The

---

[2]The challenge adopts a different group of evaluation metrics, please check their website for the formal definitions: `http://challenge.interaction-dataset.com/prediction-challenge/intro`

Table 1: INTERACTION Dataset Validation and Test Results

| Scene | Model | AR(%) | K = 1 | | K = 6 | |
|---|---|---|---|---|---|---|
| | | | $min$FDE | $min$ADE | $min$FDE | $min$ADE |
| IS | VAE | $1.85 \pm 4.02$ | $1.32 \pm 0.03$ | $0.42 \pm 0.01$ | $0.73 \pm 0.01$ | $0.26 \pm 0.01$ |
| | CVAE | $0.18 \pm 0.36$ | $\mathbf{1.27 \pm 0.02}$ | $\mathbf{0.41 \pm 0.01}$ | $\mathbf{0.68 \pm 0.02}$ | $\mathbf{0.24 \pm 0.01}$ |
| | Ours | $\mathbf{15.8 \pm 10.4}$ | $\mathbf{1.27 \pm 0.02}$ | $\mathbf{0.41 \pm 0.01}$ | $0.70 \pm 0.02$ | $0.25 \pm 0.01$ |
| RA | VAE | $0.05 \pm 0.04$ | $1.34 \pm 0.02$ | $0.42 \pm 0.01$ | $0.76 \pm 0.01$ | $0.26 \pm 0.01$ |
| | CVAE | $13.4 \pm 14.9$ | $1.32 \pm 0.01$ | $0.42 \pm 0.01$ | $0.73 \pm 0.02$ | $\mathbf{0.25 \pm 0.01}$ |
| | Ours | $\mathbf{19.5 \pm 15.4}$ | $\mathbf{1.29 \pm 0.03}$ | $0.42 \pm 0.01$ | $\mathbf{0.72 \pm 0.01}$ | $0.26 \pm 0.01$ |
| HM | VAE | $8.68 \pm 8.36$ | $0.87 \pm 0.08$ | $0.29 \pm 0.03$ | $0.42 \pm 0.04$ | $0.16 \pm 0.01$ |
| | CVAE | $5.42 \pm 10.2$ | $0.83 \pm 0.03$ | $0.28 \pm 0.03$ | $0.40 \pm 0.05$ | $0.16 \pm 0.02$ |
| | Ours | $\mathbf{15.5 \pm 8.13}$ | $\mathbf{0.65 \pm 0.09}$ | $\mathbf{0.22 \pm 0.02}$ | $\mathbf{0.32 \pm 0.04}$ | $\mathbf{0.13 \pm 0.01}$ |

| Track | Model | AR(%) | K = 6 | | | K = 50 | | |
|---|---|---|---|---|---|---|---|---|
| | | | ADE | FDE | MoN | ADE | FDE | MoN |
| R | VAE | $0.35 \pm 5.37$ | 0.5685 | 1.7573 | 0.2238 | 0.5712 | 1.7709 | 0.1036 |
| | CVAE | $0.07 \pm 1.43$ | 0.5323 | 1.6425 | 0.2144 | 0.5410 | 1.6725 | **0.0983** |
| | Ours* | $0.88 \pm 2.49$ | 0.5157 | 1.5823 | 0.2195 | 0.5158 | 1.5823 | 0.1106 |
| | Ours | $\mathbf{24.8 \pm 20.2}$ | **0.4665** | **1.4174** | **0.2011** | **0.4662** | **1.4187** | 0.1050 |
| G | VAE | $0.02 \pm 1.27$ | 1.3428 | 3.8542 | 0.8193 | 1.3436 | 3.8564 | 0.5181 |
| | CVAE | $0.05 \pm 2.20$ | 1.4517 | 4.2179 | 0.8743 | 1.3824 | 3.9972 | 0.5676 |
| | Ours* | $0.37 \pm 2.74$ | 1.1615 | 3.3927 | 0.6811 | 1.1617 | 3.3939 | **0.4218** |
| | Ours | $\mathbf{15.2 \pm 14.7}$ | **0.9205** | **2.7049** | **0.6075** | **0.9186** | **2.6969** | 0.4891 |

Ours* refers to an instance of social-CVAE that still suffers from the social posterior collapse problem. The results are presented in the format of $\mathrm{mean} \pm \mathrm{std}$. On the validation set, the mean and std are computed over multiple trials. On the test set, the mean and std of AR are computed over all the samples.

Table 2: Argoverse Dataset Validation Results

| Model | AR(%) | K = 1 | | K = 6 | |
|---|---|---|---|---|---|
| | | $min$FDE | $min$ADE | $min$FDE | $min$ADE |
| VAE | $1.27 \pm 1.00$ | $4.17 \pm 0.17$ | $1.86 \pm 0.07$ | $2.33 \pm 0.06$ | $1.30 \pm 0.04$ |
| CVAE | $0.40 \pm 0.31$ | $3.85 \pm 0.01$ | $1.73 \pm 0.01$ | $2.09 \pm 0.02$ | $1.19 \pm 0.01$ |
| Ours | $\mathbf{28.5 \pm 15.7}$ | $\mathbf{3.52 \pm 0.03}$ | $\mathbf{1.59 \pm 0.02}$ | $\mathbf{1.98 \pm 0.02}$ | $\mathbf{1.15 \pm 0.01}$ |

The results are presented in the format of $\mathrm{mean} \pm \mathrm{std}$ computed over multiple trials.

comparison between our model and other methods can be found in Appx. F. In Table 3, we see that changing to CVAE formulation leads to a significant boost in terms of AR, implying that the model gives more attention to surrounding agents. However, neither changing to CVAE nor the auxiliary task improves prediction performance. It is because historical social context is less informative for pedestrian trajectory prediction. The ETH/UCY dataset is collected in unconstrained environments with few objects. Also, compared to vehicles, pedestrians have fewer physical constraints in 2D motion, resulting in shorter temporal dependency in their movement. It is consistent with the results in [9], where the authors proposed a transformer-based model that achieves the current state-of-the-art performance on ETH/UCY. The visualized attention maps in [9] show that social context in nearby timesteps is more important to their prediction model. In contrast, the historical trajectories of other agents always receive low attention weights. Therefore, even if the auxiliary prediction task encourages our model to encode historical social context, it does not lead to either a larger AR value or better prediction performance. It suggests that the simplified latent variable model studied in this work might not be sufficient to model pedestrian motion. Explicit future interaction should be considered, and an autoregressive decoder as the one in [9] should be used to account for it.

# 6    Discussion

**Limitations.** As the first study on social posterior collapse, we cannot explore every aspect of this subject. Many possible factors could contribute to this phenomenon. Our experimental results can only speak for the particular model we develop and the tasks we study. The occurrence of social posterior collapse and its effect on the overall performance may vary across different problems and different model architectures. For instance, our finding from the pedestrian trajectory prediction task is pretty different from its vehicle counterpart. The format of the graph representation, especially how the environmental information is incorporated, also matters. Also, the diagnosis based on the AR value could be inconclusive under different graph representations. In Appx. E.2, we show that the AR values of the three models become similar after removing the lanelet nodes for the highway merging subset of the INTERACTION dataset. However, it does not mean that social posterior collapse does not occur. Our social-CVAE model can maintain consistent prediction accuracy without map information, while the baseline VAE model has a significant drop in performance. It implies that the baseline VAE model does not properly utilize historical social context, but we cannot assert that social posterior collapse occurs due to the lack of direct evidence.

Nevertheless, we do demonstrate that social posterior collapse is not unique to our sparse-GAMP module (Appx. E.1). Even if we switch to other aggregation functions, we can still find evidence suggesting its occurrence. Our sparse graph attention function, $\mathcal{G}$-entmax, is a convenient and flexible toolkit that allows us to monitor and analyze social posterior collapse without compromising performance. In the future, we will work along this direction to explore alternative diagnosis toolkits that can be applied to detect social posterior collapse for a broader range of models.

**Connections to Related Works.** We want to wrap up our discussion with a glance at those prior works on VAE-based multi-agent behavior modeling. We are curious about if any elements in their models have implicitly tackled this issue. However, we would like to emphasize that it is still necessary to explicitly study social posterior collapse under their settings in the future, which could provide helpful guidance to avoid social posterior collapse in model design. Due to the space limit, we only discuss works using techniques potentially related to social posterior collapse, especially those provided evidence that interacting behaviors were appropriately modeled (e.g., attention maps or visualized prediction results in interactive scenarios).

Trajectron [6] and Trajectron++ [4] which are formulated as CVAEs establish the previous state-of-the-art results on ETH/UCY. Different from ours, they adopted a discrete latent space to account for the multi-modality in human behavior. They did not examine how their models utilize social context. However, we think that a discrete latent space could potentially alleviate the social posterior collapse issue. Compared to a continuous random variable, a discrete one can only encode a limited amount of information. It prevents the model from bypassing social context since a discrete latent variable is not sufficient to encode all the information of a long trajectory. For the same reason, NRI [29], which adopted a discrete latent space for interacting system modeling, is also potentially relevant. PECNet [7] is another CVAE-based trajectory prediction model. Instead of conditioning on the entire future trajectory, they proposed the Endpoint VAE where the posterior latent variables only condition on the endpoints. It could also be a potential solution as it limits the amount of information the latent space can encode from the future trajectory.

In [1], Suo et al. proposed a multi-agent behavior model for traffic simulation under the CVAE framework. They demonstrated that their method was able to simulate complex and realistic interactive behaviors. In particular, they augmented ELBO with a commonsense objective that regularizes the model from synthesizing undesired interactions (e.g., collisions). We think it plays a similar role as our auxiliary prediction task. The last work we would like to discuss is the AgentFormer model [9] mentioned in Sec. 5.2. Their results suggest that incorporating an autoregressive decoder and a future social context encoder could be more effective in interaction modeling, especially for systems without long-term dependency (e.g., pedestrians). However, as we have mentioned before, the autoregressive decoder introduces another issue if it is overly powerful.

# 7    Conclusion

In this work, we point out an under-explored issue, which we refer to as social posterior collapse, in the context of VAEs in multi-agent modeling. We argue that one of the commonly adopted formulations

Table 3: ETH/UCY Dataset Leave-one-out Testing Results

| Model | ETH | | | HOTEL | | |
|---|---|---|---|---|---|---|
| | $AR(\%)$ | $min$FDE | $min$ADE | $AR(\%)$ | $min$FDE | $min$ADE |
| VAE | $74.0 \pm 7.33$ | $0.98 \pm 0.07$ | $0.63 \pm 0.03$ | $29.0 \pm 28.1$ | $0.28 \pm 0.01$ | $0.19 \pm 0.01$ |
| CVAE | $\mathbf{90.9 \pm 6.38}$ | $\mathbf{0.94 \pm 0.10}$ | $\mathbf{0.61 \pm 0.05}$ | $69.8 \pm 41.7$ | $\mathbf{0.27 \pm 0.01}$ | $\mathbf{0.18 \pm 0.01}$ |
| Ours | $83.2 \pm 12.6$ | $1.08 \pm 0.10$ | $0.68 \pm 0.04$ | $\mathbf{72.9 \pm 21.5}$ | $\mathbf{0.27 \pm 0.02}$ | $\mathbf{0.18 \pm 0.01}$ |

| Model | ZARA1 | | | ZARA2 | | |
|---|---|---|---|---|---|---|
| | $AR(\%)$ | $min$FDE | $min$ADE | $AR(\%)$ | $min$FDE | $min$ADE |
| VAE | $40.3 \pm 22.8$ | $\mathbf{0.38 \pm 0.01}$ | $\mathbf{0.22 \pm 0.01}$ | $30.1 \pm 17.1$ | $\mathbf{0.32 \pm 0.01}$ | $\mathbf{0.18 \pm 0.01}$ |
| CVAE | $\mathbf{89.4 \pm 7.65}$ | $0.39 \pm 0.01$ | $\mathbf{0.22 \pm 0.01}$ | $\mathbf{64.2 \pm 25.6}$ | $0.35 \pm 0.01$ | $0.19 \pm 0.01$ |
| Ours | $61.9 \pm 5.71$ | $\mathbf{0.38 \pm 0.01}$ | $\mathbf{0.22 \pm 0.01}$ | $58.6 \pm 15.7$ | $0.37 \pm 0.02$ | $0.20 \pm 0.01$ |

| Model | UNIV | | | Average | | |
|---|---|---|---|---|---|---|
| | $AR(\%)$ | $min$FDE | $min$ADE | $AR(\%)$ | $min$FDE | $min$ADE |
| VAE | $0.06 \pm 0.09$ | $\mathbf{0.55 \pm 0.01}$ | $\mathbf{0.31 \pm 0.01}$ | $34.7 \pm 29.4$ | $\mathbf{0.50 \pm 0.27}$ | $\mathbf{0.30 \pm 0.17}$ |
| CVAE | $\mathbf{41.2 \pm 25.1}$ | $0.63 \pm 0.05$ | $0.35 \pm 0.02$ | $\mathbf{71.1 \pm 29.5}$ | $0.51 \pm 0.25$ | $0.31 \pm 0.17$ |
| Ours | $36.3 \pm 14.4$ | $0.63 \pm 0.02$ | $0.35 \pm 0.01$ | $62.6 \pm 21.0$ | $0.54 \pm 0.29$ | $0.33 \pm 0.19$ |

The results are presented in the format of $\mathrm{mean} \pm \mathrm{std}$ computed over multiple trials.

of VAEs in multi-agent modeling is prone to ignoring historical social context when predicting the future trajectory of an agent. We analyze the reason behind and propose several measures to alleviate social posterior collapse. Afterward, we design a GNN-based realization of the general framework incorporating the proposed measures, which we refer to as social-CVAE. In particular, social-CVAE incorporates a novel sparse-GAMP layer which helps us detect and analyze social posterior collapse. In our experiments, we show that social posterior collapse occurs in real-world trajectory prediction problems and that the proposed measures effectively alleviate the issue. Also, the experimental results imply that social posterior collapse could cause poor generalization performance in novel scenarios if the future movement of the agents indeed depends on their historical social context. In the future, we will utilize the toolkit developed in this work to explore social posterior collapse in a broader range of model architectures and interacting systems.

## Acknowledgements

We would like to thank Liting Sun, Xiaosong Jia, and Jiachen Li for insightful discussion. We thank Vade Shah for helping us with the experiments. We thank Chenfeng Xu and Hengbo Ma for their valuable feedback. We would also like to thank the anonymous reviewers for their helpful review comments. This work is supported by Denso International America, Inc.

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
