## A  Proof for Proposition 1

*Proof.* Because the graph in Fig. 1 is a directed acyclic graph (DAG), we can apply the $d$-separation criterion [43] to analyze the conditional independence. For each pair of $i$ and $j$ with $i \neq j$, the node $\mathbf{z}_i$ is connected to $\mathbf{y}_j$ through $\mathbf{T}_i$ and $\mathbf{T}_j$. And any path between $\mathbf{T}_i$ and $\mathbf{T}_j$ has a triple $\mathbf{T}_i \leftarrow \mathbf{x}_k \rightarrow \mathbf{T}_j$ for some $k = 1, 2, ..., n$, which is inactive after conditioning on $\mathbf{x}$. Therefore, we can apply the $d$-separation criterion to conclude that $(\mathbf{z}_i \perp\!\!\!\perp \mathbf{y}_j) \mid \mathbf{x}, \mathbf{y}_i$. The same inactive triples also imply that $(\mathbf{z}_i \perp\!\!\!\perp \mathbf{z}_j) \mid \mathbf{x}$. For the third statement, any path between $\mathbf{z}_i$ and $\mathbf{x}_j$ has a inactive path $\mathbf{x}_j \rightarrow \mathbf{T}_i \rightarrow \mathbf{y}_i$. Therefore, the $d$-separation criterion implies $(\mathbf{z}_i \perp\!\!\!\perp \mathbf{x}_j) \mid \mathbf{T}_i$. $\square$

## B  Proof for Proposition 2

*Proof.* Proposition 3 in [22] suggests that the threshold value equals to $\tau_{\mathbf{s}/2}(\rho^*)$ with any $\rho^*$ satisfying $\tau_{\mathbf{s}/2}(\rho^*) \in [\frac{s_{[\rho^*+1]}}{2}, \frac{s_{[\rho^*]}}{2}]$. If $d$ satisfies the condition, then $\tau_{\mathbf{s}/2}(d) \in (-\infty, \frac{s_{[d]}}{2}]$ which is clearly finite. Therefore, $\tau_{\mathbf{s}/2}(d) = M_{\mathbf{s}/2}(d) - \sqrt{\frac{1 - S_{\mathbf{s}/2}(d)}{d}} \geqslant \frac{s_{[d]}}{2} - 1$ by definition. Given $\frac{s'_{d+1}}{2} \leqslant \frac{s_{[d]}}{2} - 1$, we have $\tau_{\mathbf{s}'/2}(d) = \tau_{\mathbf{s}/2}(d) \in [\frac{s'_{d+1}}{2}, \frac{s_{[d]}}{2}]$. Therefore, $\tau_{\mathbf{s}/2}(d)$ is still the threshold value. If $d$ is not a valid $\rho^*$, then there exists $\rho^* \in \{1, ..., d-1\}$ defining the threshold value. Because $\frac{s'_{d+1}}{2} < \frac{s_{[d]}}{2}$, augmenting the input vector does not affect the threshold value. In both cases, the threshold value is unchanged which leads to the first statement of the proposition.

Now we prove the second statement of the proposition. Because $p_i > 0$ for $i = 1, .., d$, the threshold value is smaller than $\frac{s_{[d]}}{2}$, which makes $d$ the only possible $\rho^*$. If $s'_{d+1} \leqslant 2\tau_{\mathbf{s}/2}(d)$, then $\tau_{\mathbf{s}/2}(d)$ defines the threshold value for $\mathbf{s}'$. Therefore, $p'_i = p_i$ for $i = 1, 2, ..., d$ and $p'_{d+1} = 0$. Instead, if we are given $p'_i = p_i$ for $i = 1, 2, ..., d$ and $p'_{d+1} = 0$, the threshold satisfies $\tau_{\mathbf{s}'/2}(\rho^*) \in \left[ \frac{s'_{d+1}}{2}, \frac{s_{[d]}}{2} \right]$. Therefore, $d$ is a valid threshold index for both $\mathbf{s}$ and $\mathbf{s}'$, and $s'_{d+1} \leqslant 2\tau_{\mathbf{s}'/2}(d) = 2\tau_{\mathbf{s}/2}(d)$. $\square$

## C  Experiment Details

In this section, we report additional experiment details for the experiments in Sec. 5, including data processing scheme, implementation details, and hyper-parameters used.

### C.1  Data Processing Scheme

**INTERACTION Dataset.** The access to the dataset is granted for non-commercial usage through its official website: `https://interaction-dataset.com` (Copyright©All Rights Reserved). First, we use the script from the official code repository to split the training and validation sets and segment the data. Each sample has a length of 4 seconds, with an observation window of 1 second and a prediction window of 3 seconds. The sampling frequency is 10Hz. We further augment the dataset by iteratively assigning each vehicle in a sample as the target vehicle. Afterward, we follow the common practice to translate and rotate the coordinate system, such that the target vehicle locates at the origin with a zero-degree heading angle at the last observed frame.

Regarding map information, the INTERACTION dataset provides maps in a format compatible with the lanelet representation. For each lanelet, we fit its boundaries to two B-splines through spline regression so that we can uniformly sample a fixed number of points on each boundary. Apart from the coordinates, we add additional discrete features (e.g., boundary type, the existence of a stop sign) to each boundary point.

**Argoverse Dataset.** We download the dataset (v1.1) including the training, validation and testing subsets from its official website: `https://www.argoverse.org/data.html` (Copyright©2020 Argo AI, LLC). Each sample in the Argoverse Dataset has a length of 5 seconds, with an observation window of 2 seconds and a prediction window of 3 seconds. The sampling frequency is 10Hz. We follow a similar scheme to process the dataset. However, since each sample contains vehicles located in many city blocks, we first filter out those irrelevant vehicles and road segments. We remove surrounding vehicles whose distance to the target vehicle is larger than a certain threshold value at the last observed frame. To identify irrelevant road segments, we use a heuristic-based graph search

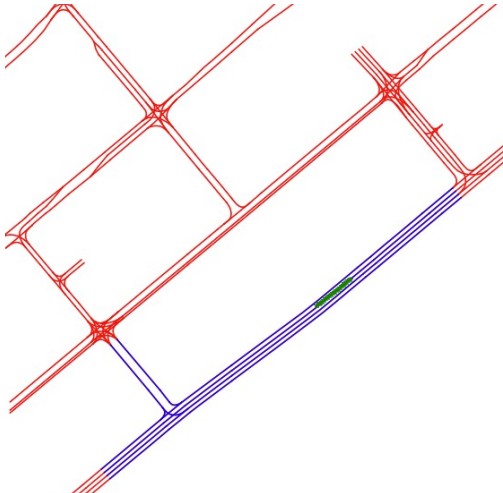

Figure 3: Road segments found by the graph search algorithm. We denote the observed trajectory of the target vehicle by the green dots. The road segments returned by the graph search algorithm are highlighted in the color of blue, whereas the other road segments are drawn in the color of red.

algorithm similar to the one used in [32] to obtain the road segments of interest (ROI). Instead of setting a threshold Euclidean distance, we decide whether a road segment is relevant by estimating the traveling distance from the target vehicle's location to it. The algorithm is summarized in Alg. 1. Given $x$, the coordinates of the target vehicle at the last observed frame, we set the traveling distance threshold $d_{\max}$ based on the displacement of the target vehicle during the observed time horizon. A larger distance threshold is necessary if the target vehicle is driving at high speed. We initialize the graph search by finding road segments close to $x$ from the map. Afterward, we expand the searching graph by adding adjacent, preceding, and succeeding road segments and finding all segments within the threshold of traveling distances. We use simple heuristics to compute the traveling distance between two segments. If two segments are adjacent, the distance is set to zero. If one segment is a predecessor or successor of the other segment, we set the distance to be the average of their centerlines' length. Fig. 3 shows an example of the resulting road segments.

Another issue is that the heading angles of the vehicles are not provided. We need to estimate the heading angles of the target vehicles in order to rotate the coordinate system. However, the trajectory data are noisy with tracking errors. Simply interpolating the coordinates between consecutive time steps results in noisy estimation. Instead, we first estimate the heading angle at each observed frame by interpolating the coordinates and then obtain a smooth estimation of the heading angle at the last observed frame as follows:

$$\hat{\psi}_{T_h} = \sum_{t=0}^{T_h} \lambda^{T_h - t} \psi_t,$$

where $T_h$ is the number of observed frames, $\psi_t$ is the estimated heading at the $t^{\text{th}}$ frame, $\lambda \in (0, 1)$ is the forgetting factor, and $\hat{\psi}_{T_h}$ is the smoothed heading estimation at the last observed frame.

**ETH/UCY Dataset.** For the ETH/UCY datasets, we adopt the leave-one-out evaluation protocol as in prior works [2, 42, 4, 7, 9] to obtain five groups of datasets: ETH & HOTEL (from ETH) and UNIV, ZARA1 & ZARA2 (from UCY). The data from the corresponding scenario are left out as testing data in each group, and the remaining data are used for training and validation. We used the segmented datasets provided by the code base of social-GAN (MIT License) [2]. Each sample has a length of 8 seconds, with an observation window of 3.2 seconds and a prediction window of 4.8 seconds. The sampling frequency is 2.5Hz. We do not use any visual or semantic information for fair comparisons to prior works. The origin of the coordinate system is translated to the mean position of all agents at the last observed frame. Random rotation [42, 4] is adopted for data augmentation.

Algorithm 1: ROI Graph Search

1: **function** ROIGRAPHSEARCH($x, d_{\max}, d_{\text{init}}, map$)
2:     $segments \leftarrow map.\text{segmentsContainXY}(x)$                                   ▷ A set of segments containing $x$
3:     **while** $segments$ is empty and $d_{\text{init}} < d_{\max}$ **do**                     ▷ Search local region if empty
4:         $segments \leftarrow map.\text{segmentsInBoundingBox}(x, d_{\text{init}})$
5:         $d_{\text{init}} \leftarrow 2d_{\text{init}}$
6:     **end while**
7:     $pool \leftarrow$ initialize a FIFO queue
8:     **for** $segment$ in $segments$ **do**
9:         $node \leftarrow \text{Node}(segment, 0, 0)$                    ▷ Create node with segment, length and distance
10:         $pool \leftarrow \text{Insert}(node, pool)$
11:     **end for**
12:     $ROI \leftarrow$ an empty dictionary of nodes
13:     **while** $pool$ is not empty **do**
14:         $node \leftarrow \text{Pop}(pool)$
15:         **if** $node.segment$ not in $ROI$ or $ROI[segment].distance \geqslant node.distance$ **then**
16:             $ROI[segment] \leftarrow node$
17:         **end if**
18:         $children \leftarrow map.\text{adjacentSegments}(node.segment)$     ▷ Get adjacent road segments
19:         **for** $child$ in $children$ **do**
20:             $length \leftarrow map.\text{segmentCenterline}(child)$                     ▷ Get centerline length
21:             $node \leftarrow \text{Node}(child, length, node.distance)$
22:             $pool \leftarrow \text{Insert}(node, pool)$
23:         **end for**
24:         $predecessors \leftarrow map.\text{predecessor}(node.segment)$     ▷ Get preceding road segments
25:         $successors \leftarrow map.\text{successor}(node.segment)$       ▷ Get succeeding road segments
26:         $children \leftarrow predecessors + successors$       ▷ Combine predecessors and successors
27:         **for** $child$ in $children$ **do**
28:             $length \leftarrow map.\text{segmentCenterline}(child)$
29:             $distance \leftarrow \frac{1}{2}(length + node.length) + node.distance$
30:             **if** $distance \leqslant d_{\max}$ **then**
31:                 $node \leftarrow \text{Node}(child, length, distance)$
32:                 $pool \leftarrow \text{Insert}(node, pool)$
33:             **end if**
34:         **end for**
35:     **end while**
36:     **return** $ROI$
37: **end function**

## C.2   Implementation Details

We implement our social-CVAE model using Pytorch 1.8 (Copyright©Facebook, Inc) [26] and Pytorch Geometric (PyG) 1.7 (MIT License) [44], a geometric deep learning extension library for Pytorch which implements various popular GNN models. The sparse-GAMP network is implemented by adapting the base message-passing class from PyG. The $\mathcal{G}$-entmax function is implemented using the entmax package (MIT License) from [22]. The function $f_e$ in Eqn. (3) consists of two MLP networks: one network encodes $\mathbf{h}_i$ and $\mathbf{h}_j$ separately; one network generates $\mathbf{h}_{(i,j)}$ after concatenating the node embeddings with $\mathbf{u}_{(i,j)}$. We select the hyper-parameters through cross-validation with the aid of Tune (Copyright©2021 The Ray Team) [45]. All the MLPs used in the model are one-layer MLPs with layer normalization applied [46]. All the networks, including MLPs and GRUs, have 64 hidden units. For the experiments on the INTERACTION dataset, we choose a latent space of 16 dimensions. For the experiments on the Argoverse dataset and the ETH/UCY dataset, we choose a latent space of 32 dimensions. Regarding the loss function, we follow the common practice to fix the variance of $p_{\theta,\phi}(\mathbf{y}|\mathbf{x}, \mathbf{z})$. Consequently, the reconstruction loss is equivalent to an MSE loss, and the

overall objective becomes maximizing the following objective function:

$$\hat{\mathcal{L}}(\theta, \psi, \phi, \eta, \zeta) = \mathbb{E}_{\mathbf{x}, \mathbf{y} \sim D} \left[ \mathbb{E}_{\mathbf{z} \sim q_{\theta, \psi}(\mathbf{z} | \mathbf{x}, \mathbf{y})} \left\| h_{\theta, \phi}(\mathbf{x}, \mathbf{z}) - \mathbf{y} \right\|^2 - \beta D_{KL} \left[ q_{\theta, \psi}(\mathbf{z} | \mathbf{x}, \mathbf{y}) \| p_{\theta, \eta}(\mathbf{z} | \mathbf{x}) \right] \right]$$
$$- \alpha \mathbb{E}_{\mathbf{x}, \mathbf{y} \sim D, \mathbf{z} \sim p_{\theta, \eta}(\mathbf{z} | \mathbf{x})} \left\| h_{\theta, \zeta}(\mathbf{x}, \mathbf{z}) - \mathbf{y} \right\|^2.$$

For the vehicle prediction experiments, we set $\beta = 0.03$. For the pedestrian prediction experiments, we set $\beta = 0.01$. For the weight of the auxiliary loss function, we set $\alpha = 0.3$ on the INTERACTION dataset, $\alpha = 0.5$ on the Argoverse dataset, and $\alpha = 0.2$ on the ETH/UCY dataset. We use Adam [47] to optimize the objective function. To generate trajectories for evaluation, if $K = 1$, we sample a single trajectory by taking the mean from the prior or conditional prior as the latent variables. If $K > 1$, we randomly sample the latent variables to generate multiple trajectories. However, the Argoverse Motion Forecasting Challenge evaluates the metrics for $K = 1$ by picking one trajectory out of all the submitted samples. Therefore, for evaluation on the Argoverse dataset, we randomly sample $K - 1$ trajectories and leave the last one as the trajectory corresponding to the mean value of the latent variables. For the vehicle prediction experiments, all the models are trained for 100 epochs with a batch size of 40. For the pedestrian experiments, we choose a batch size of 20. All the models were trained with four RTX 2080 Ti and an Intel Core i9-9920X (12 Cores, 3.50 GHz). However, only a quarter of a single GPU is required to train one model.

## D    Visualization

In this section, we visualize the experiment results for the vehicle prediction task. Fig. 4 and Fig. 5 show several examples from the INTERACTION dataset and the Argoverse dataset respectively. For each instance, we compare the outputs of the three model variants we study in Sec. 5. The target vehicles' historical trajectories and ground-truth future trajectories are denoted by blue and red dots, respectively. We sample six predicted trajectories from each model and plot them in dash lines of different colors. We also approximate the density function of the prediction output using a kernel density estimator and visualize it with a color map. Since we are particularly interested in monitoring the social posterior collapse phenomenon, we visualize the attention map by highlighting the surrounding vehicles and lanelets that receive non-zero attention weights. Particularly, if a vehicle received non-zero attention, we use the same way as the target vehicle to annotate its historical and future trajectories. Otherwise, its trajectories are annotated with grey dots. If a lanelet node receives non-zero attention, we highlight its boundaries or centerline with orange lines. The VAE and CVAE variants ignore all the surrounding vehicles in all the visualized instances, including those close to the target vehicles. They only pay attention to the lanelet nodes, which results in insensible prediction results, for instance, colliding into preceding vehicles (e.g., the second row in Fig. 4 and Fig. 5). In contrast, the social-CVAE models assign attention weights to vehicles that might potentially interact with the target vehicles and maintain sparse attention maps in dense traffic scenes (e.g., the last two rows in Fig. 4).

## E    Ablation Study

### E.1    Effect of Aggregation Functions

In this section, we present an ablation study on the aggregation functions used in the message-passing network. Because of the sparsity nature of $\alpha$-entmax, the usage of sparse graph attention may induce social posterior collapse. Therefore, we would like to study whether social posterior collapse will still occur if we switch to other aggregation functions. We consider two variants for comparison. The first one is replacing $\mathcal{G}$-entmax with the conventional softmax function, resulting in the following message-passing operations:

$$v \to e: \ \mathbf{h}_{(i,j)} = f_e \left( \left[ \mathbf{h}_i, \mathbf{h}_j, \mathbf{u}_{(i,j)} \right] \right),$$
$$e \to v: \quad \hat{\mathbf{h}}_j = \sum_{i \in \mathcal{N}_j} w_{(i,j)} \mathbf{h}_{(i,j)}, \ \text{ where } \mathbf{w}_j = \text{softmax} \left( \left\{ \mathbf{h}_{(i,j)} \right\}_{i \in \mathcal{N}_j} \right),$$

The second variant is using the $\max$ aggregation instead of the weighted sum. The message-passing layer becomes the one in Eqn. (4) - (5). The $\max$ aggregation takes the element-wise maximum along the dimension of the hidden unit. Therefore, it allows the number of activated nodes to be

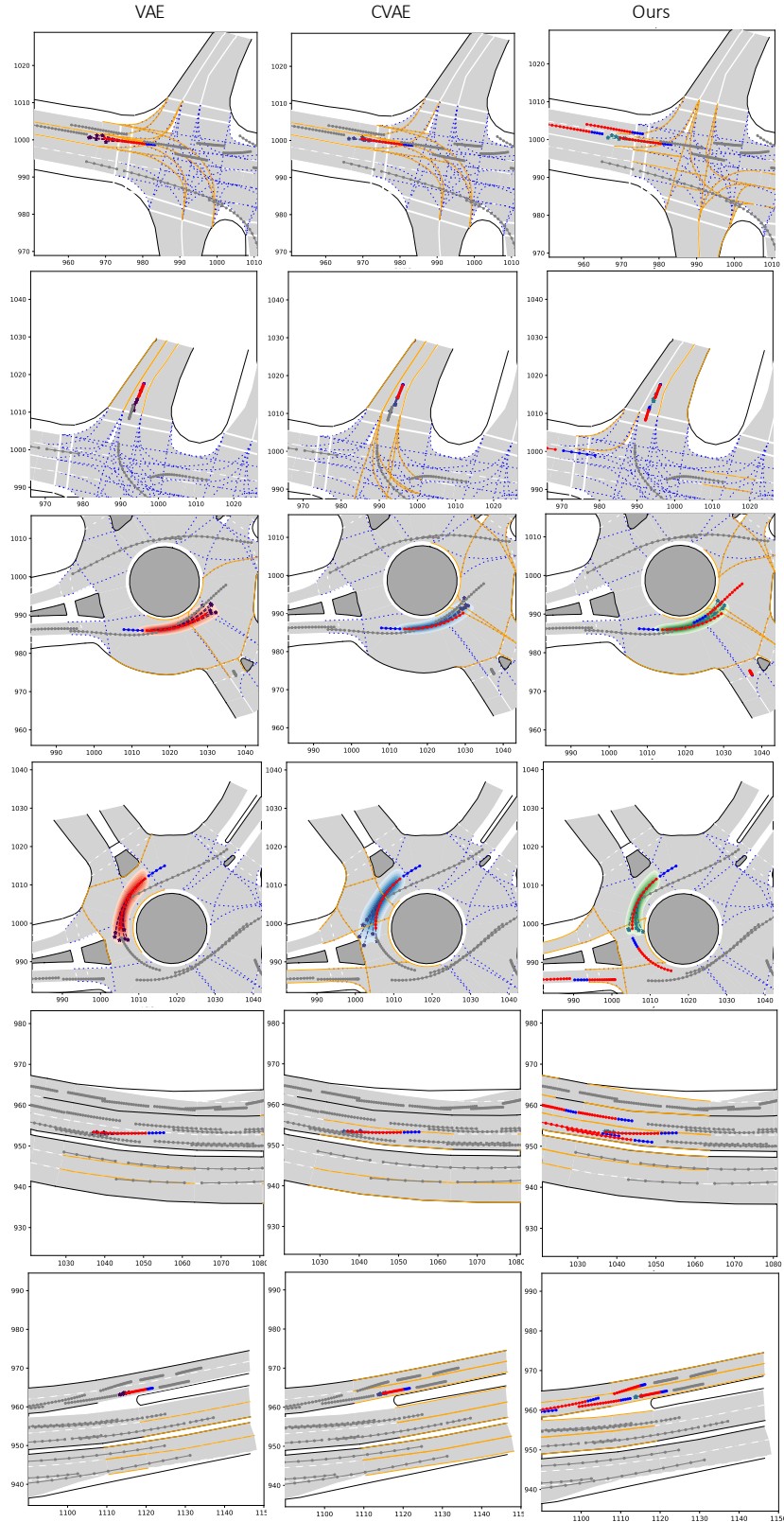

Figure 4: Visualizing prediction results on the Interaction dataset.

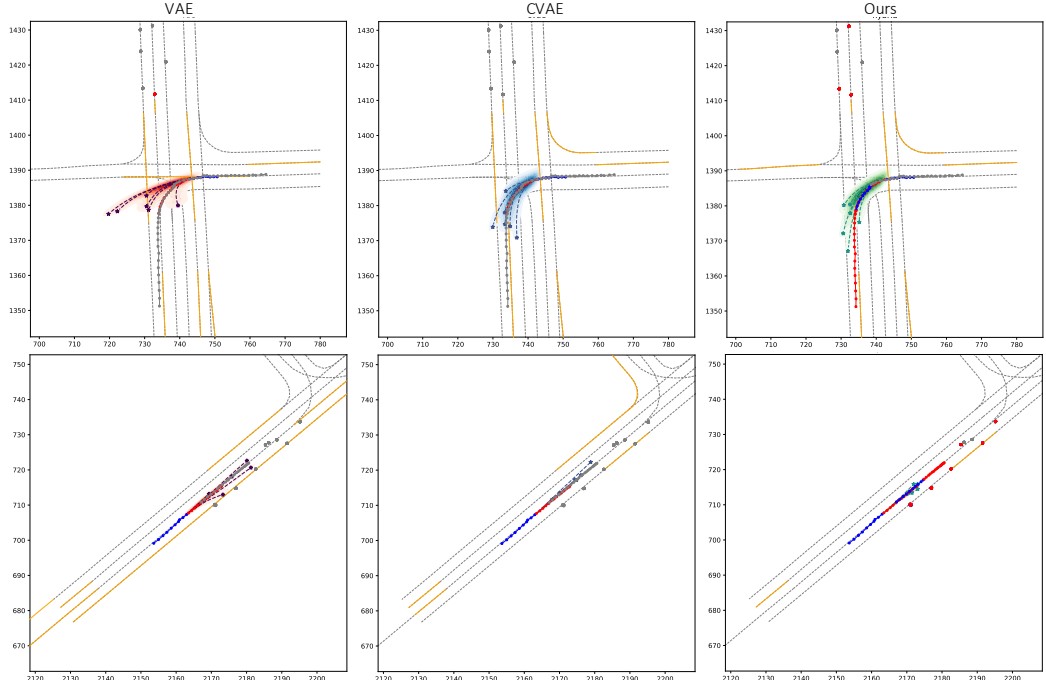

Figure 5: Visualizing prediction results on the Argoverse dataset.

at most the dimension of $\mathbf{h}_{(i,j)}$. The social posterior collapse issue should be able to be avoided if $\mathcal{G}$-entmax is the reason for its occurrence.

$$v \rightarrow e: \mathbf{h}_{(i,j)} = f_e\left(\left[\mathbf{h}_i, \mathbf{h}_j, \mathbf{u}_{(i,j)}\right]\right), \tag{4}$$

$$e \rightarrow v: \quad \hat{\mathbf{h}}_j = \text{max-aggregate}\left(\{\mathbf{h}_{(i,j)}\}_{i \in N_j}\right). \tag{5}$$

The issue that remains is the evaluation metric. Unlike sparse-GAMP, we do not have the privilege to detect social posterior collapse by monitoring the magnitude of AR. We need to find alternative and unified evaluation metrics to compare models with different aggregation functions. We adopt the two feature importance measures used in [36] as our evaluation metrics: 1) gradient-based measures of feature importance, and 2) differences in model output induced by leaving features out. However, instead of studying the importance of single features, we are interested in the contribution of a single agent to model output. Consequently, we define a customized gradient-based measure as follows:

$$\tau_{g,i} = \frac{1}{2(n-1)T_h} \sum_{j=1, j \neq i}^{n} \left\|\frac{\partial \hat{\mathbf{y}}_{i,T_p}}{\partial \mathbf{x}_j}\right\|_{1,1},$$

where $i$ is the index of the target agent, $T_p$ is the number of predicted frames, $\hat{\mathbf{y}}_{i,T_p}$ is the predicted state of the target agent at the last frame, $\partial \hat{\mathbf{y}}_{i,T_p}/\partial \mathbf{x}_j$ is the partial Jacobian matrix of $\hat{\mathbf{y}}_{i,T_p}$ regarding the observed trajectory of the agent $j$. And $\|\cdot\|_{1,1}$ defines the entry-wise 1-norm which sums the absolute values of the matrix's entries. $\tau_{g,i}$ essentially measures the average gradient of the model output regarding the observations of the surrounding agents. If the model ignores social context, a small $\tau_{g,i}$ is expected. Also, we define a customized leave-one-out ADE metric as follows:

$$loo\text{ADE}_i = \frac{1}{n-1} \sum_{j=1, j \neq i}^{n} \text{ADE}\left(\hat{\mathbf{y}}(\mathbf{x}_{-j}), \hat{\mathbf{y}}(\mathbf{x})\right),$$

where $\mathbf{x}_{-j}$ denotes the observed trajectories with the agent $j$ masked out. $loo\text{ADE}_i$ equals to the average ADE between the normal prediction output and the prediction output with each surrounding agent masked out. Similar to $\tau_{g,i}$, we expect a small $loo\text{ADE}_i$ if social posterior collapse occurs.

We conduct the ablation study on the vehicle prediction task. On both datasets, we repeated the experiments but replaced the models with their variants with different aggregation functions. We are mainly interested in comparing the values of $\tau_g$ and $loo$ADE across the models with the same aggregation functions. Because the other aggregation functions do not encourage a sparse model structure, any surrounding agent could contribute to $\tau_g$ and $loo$ADE. It makes it less informative to compare them against the model with sparse attention. The results on the validation sets are summarized in Table 4 and 5. The trends in $\tau_g$ and $loo$ADE are similar regardless of the aggregation functions. In most circumstances, our social-CVAE model attains the highest scores with large margins, whereas the VAE variant has the lowest $\tau_g$ and $loo$ADE. Meanwhile, the comparisons in prediction performance are also consistent with the case of sparse-GAMP. Therefore, we can conclude that social posterior collapse is not unique to the models with sparse-GAMPs.

We also observe that changing the formulation from VAE to CVAE always leads to an increase in $\tau_g$ and $loo$ADE when the alternative aggregation functions are used. In contrast, the values could stay unchanged in the case of sparse-GAMP. We are then curious if merely changing the formulation can alleviate social posterior collapse when sparse-GAMP is not used. We take the softmax variant as an example and investigate its attention maps. However, simply computing the ratio of non-zero attention weights is meaningless because the attention is no longer sparse. To solve this problem, we refine our definition of AR as follows:

$$\text{AR}_\delta = \frac{\sum_{i=1, i \neq j}^{n} \mathbf{1}(\omega_{(i,j)} \geqslant \delta)}{n - 1}, \quad \delta \in [0, 1].$$

Now $\text{AR}_\delta$ is a function of a threshold value $\delta$. Instead of looking at a single value at $\delta = 0$, we are interested in seeing how $\text{AR}_\delta$ changes when increasing $\delta$ from 0 to 1. In Fig. 6, we plot $\text{AR}_\delta(\%)$ versus the threshold $\delta$ for all the models with softmax and entmax functions on the Argoverse dataset. By switching to the conditional model, the softmax variant assigns relatively larger attention weights to surrounding agents. However, the increase in $\text{AR}_\delta$ mainly occurs at small threshold values. Compared to the social-CVAE models, the ratio of agents receiving large attention weights is lower. It is consistent with our argument that merely changing the formulation is insufficient to alleviate social posterior collapse. Another interesting observation is that the $\text{AR}_\delta$ curves for the two variants of social-CVAE models coincide when $\delta \geqslant 0.1$. It implies that our sparse graph attention does not prevent the model from identifying interacting agents. It just filters out agents that are recognized as irrelevant to maintain a sparse and interpretable attention map. We can also observe from the evaluated prediction metrics that the sparse graph attention does not interfere with the prediction performance. The social-CVAE models with either attention mechanism achieve similar prediction accuracy. In short, our sparse graph attention function provides a convenient and flexible toolkit that allows us to monitor and analyze social posterior collapse without compromising performance. Although we can still analyze the models without it, these universal metrics, i.e., $\tau_g$ and $loo$ADE, are computationally expensive, especially for evaluation at run time.

## E.2   Effect of Environmental Information

In this section, we investigate the effect of environmental information on social posterior collapse. In Sec. 5, we find the models behave quite differently on the pedestrian prediction task and its vehicle counterpart. In our experiments on the ETH/UCY dataset, although switching to CVAE leads to a larger AR value, it does not boost prediction performance. Moreover, the auxiliary task neither improves prediction performance nor further increases the AR value. In the main text, we argue that it is because, unlike vehicles, pedestrians do not have a long-term dependency on their interaction. As a result, the model cannot benefit from encoding historical social context. However, we also adopt different input representations for the two prediction problems. In the vehicle prediction task, the input graphs have additional lanelet nodes, contributing to the difference in model behavior.

To answer this question, we trained models for vehicle prediction with the same representation as pedestrians, which means removing lanelet nodes and adding self-edges instead. We choose to study this problem on the highway merging subset of the INTERACTION dataset because the interaction between vehicles depends less on the map than urban driving. We follow the same practice to run five trials for each model variant and report all the metrics' mean and standard deviation. The results are summarized in Table 6. We also copy the results from Table 1 for convenient comparison.

We think social posterior collapse still occurs in the baseline VAE model. As social context affects highway driving to a larger degree than road structure, our social-CVAE model that encodes social

Table 4: INTERACTION Dataset Aggregation Function Comparison

| Scene | Aggreg. | Model | $\tau_g$ $(\times 10^{-4})$ | $loo$ADE $(\times 10^{-3})$ | K = 6 | |
|---|---|---|---|---|---|---|
| | | | | | $min$ADE | $min$FDE |
| IS | max | VAE | $0.39 \pm 0.10$ | $0.52 \pm 0.20$ | $0.28 \pm 0.01$ | $0.79 \pm 0.01$ |
| | | CVAE | $3.86 \pm 1.30$ | $6.38 \pm 2.76$ | $\mathbf{0.24 \pm 0.01}$ | $\mathbf{0.69 \pm 0.01}$ |
| | | Ours | $\mathbf{14.8 \pm 5.39}$ | $\mathbf{20.3 \pm 3.25}$ | $0.25 \pm 0.01$ | $0.70 \pm 0.01$ |
| | softmax | VAE | $0.90 \pm 1.37$ | $1.41 \pm 2.35$ | $0.26 \pm 0.01$ | $0.73 \pm 0.02$ |
| | | CVAE | $5.00 \pm 1.04$ | $10.6 \pm 2.16$ | $\mathbf{0.24 \pm 0.01}$ | $\mathbf{0.67 \pm 0.01}$ |
| | | Ours | $\mathbf{33.6 \pm 19.0}$ | $\mathbf{17.7 \pm 1.37}$ | $0.25 \pm 0.02$ | $0.70 \pm 0.01$ |
| | entmax | VAE | $0.05 \pm 0.09$ | $0.20 \pm 0.42$ | $0.26 \pm 0.01$ | $0.73 \pm 0.02$ |
| | | CVAE | $0.02 \pm 0.03$ | $0.02 \pm 0.03$ | $\mathbf{0.24 \pm 0.01}$ | $\mathbf{0.68 \pm 0.01}$ |
| | | Ours | $\mathbf{29.6 \pm 2.50}$ | $\mathbf{8.73 \pm 5.50}$ | $0.25 \pm 0.01$ | $0.70 \pm 0.02$ |
| RA | max | VAE | $0.15 \pm 0.05$ | $0.36 \pm 0.17$ | $0.30 \pm 0.01$ | $0.86 \pm 0.02$ |
| | | CVAE | $5.12 \pm 2.18$ | $8.42 \pm 2.41$ | $\mathbf{0.26 \pm 0.01}$ | $\mathbf{0.74 \pm 0.01}$ |
| | | Ours | $\mathbf{36.3 \pm 12.4}$ | $\mathbf{28.8 \pm 1.40}$ | $0.28 \pm 0.01$ | $0.76 \pm 0.02$ |
| | softmax | VAE | $1.62 \pm 1.57$ | $4.51 \pm 3.30$ | $0.27 \pm 0.01$ | $0.79 \pm 0.02$ |
| | | CVAE | $8.63 \pm 2.85$ | $18.4 \pm 1.18$ | $\mathbf{0.26 \pm 0.01}$ | $\mathbf{0.73 \pm 0.01}$ |
| | | Ours | $\mathbf{40.6 \pm 10.2}$ | $\mathbf{22.8 \pm 0.74}$ | $0.27 \pm 0.01$ | $\mathbf{0.73 \pm 0.01}$ |
| | entmax | VAE | $0.00 \pm 0.00$ | $0.01 \pm 0.01$ | $0.26 \pm 0.01$ | $0.75 \pm 0.01$ |
| | | CVAE | $11.8 \pm 10.9$ | $5.91 \pm 5.70$ | $\mathbf{0.25 \pm 0.01}$ | $\mathbf{0.72 \pm 0.01}$ |
| | | Ours | $\mathbf{71.0 \pm 71.4}$ | $\mathbf{12.9 \pm 8.64}$ | $0.26 \pm 0.01$ | $0.73 \pm 0.02$ |
| HM | max | VAE | $0.45 \pm 0.36$ | $0.40 \pm 0.18$ | $0.20 \pm 0.02$ | $0.52 \pm 0.05$ |
| | | CVAE | $37.8 \pm 13.3$ | $7.00 \pm 0.68$ | $0.15 \pm 0.01$ | $0.36 \pm 0.01$ |
| | | Ours | $\mathbf{196 \pm 53.0}$ | $\mathbf{8.28 \pm 0.57}$ | $\mathbf{0.13 \pm 0.01}$ | $\mathbf{0.31 \pm 0.01}$ |
| | softmax | VAE | $8.89 \pm 0.69$ | $3.92 \pm 2.68$ | $0.17 \pm 0.01$ | $0.42 \pm 0.02$ |
| | | CVAE | $47.0 \pm 8.01$ | $\mathbf{11.5 \pm 0.53}$ | $0.15 \pm 0.01$ | $0.34 \pm 0.01$ |
| | | Ours | $\mathbf{176 \pm 36.0}$ | $9.42 \pm 0.40$ | $\mathbf{0.14 \pm 0.01}$ | $\mathbf{0.32 \pm 0.01}$ |
| | entmax | VAE | $10.7 \pm 22.4$ | $1.80 \pm 3.36$ | $0.16 \pm 0.01$ | $0.41 \pm 0.04$ |
| | | CVAE | $16.4 \pm 36.6$ | $1.93 \pm 4.14$ | $0.15 \pm 0.02$ | $0.38 \pm 0.04$ |
| | | Ours | $\mathbf{205 \pm 158}$ | $\mathbf{6.23 \pm 3.27}$ | $\mathbf{0.13 \pm 0.02}$ | $\mathbf{0.32 \pm 0.04}$ |

Table 5: Argoverse Dataset Aggregation Function Comparison

| Aggreg. | Model | $\tau_g$ $(\times 10^{-3})$ | $loo$ADE $(\times 10^{-2})$ | K = 6 | |
|---|---|---|---|---|---|
| | | | | $min$ADE | $min$FDE |
| max | VAE | $0.11 \pm 0.12$ | $0.24 \pm 0.31$ | $1.47 \pm 0.09$ | $2.64 \pm 0.07$ |
| | CVAE | $3.91 \pm 2.06$ | $3.33 \pm 1.82$ | $1.22 \pm 0.02$ | $2.15 \pm 0.05$ |
| | Ours | $\mathbf{13.5 \pm 2.35}$ | $\mathbf{13.9 \pm 0.77}$ | $\mathbf{1.18 \pm 0.04}$ | $\mathbf{2.07 \pm 0.09}$ |
| softmax | VAE | $0.12 \pm 0.06$ | $0.29 \pm 0.12$ | $1.36 \pm 0.11$ | $2.40 \pm 0.07$ |
| | CVAE | $5.85 \pm 1.64$ | $5.62 \pm 1.00$ | $1.16 \pm 0.01$ | $1.98 \pm 0.01$ |
| | Ours | $\mathbf{13.1 \pm 2.17}$ | $\mathbf{8.83 \pm 1.27}$ | $\mathbf{1.13 \pm 0.01}$ | $\mathbf{1.95 \pm 0.03}$ |
| entmax | VAE | $0.02 \pm 0.02$ | $0.07 \pm 0.06$ | $1.29 \pm 0.04$ | $2.32 \pm 0.05$ |
| | CVAE | $0.01 \pm 0.01$ | $0.02 \pm 0.01$ | $1.19 \pm 0.01$ | $2.08 \pm 0.02$ |
| | Ours | $\mathbf{7.30 \pm 2.95}$ | $\mathbf{5.77 \pm 2.34}$ | $\mathbf{1.15 \pm 0.02}$ | $\mathbf{1.97 \pm 0.02}$ |

context can maintain consistent prediction accuracy without map information. On the other hand, removing lanelet nodes leads to a significant drop in the baseline VAE model's prediction accuracy. It indicates that the baseline VAE model does not utilize social context well but relies heavily on map

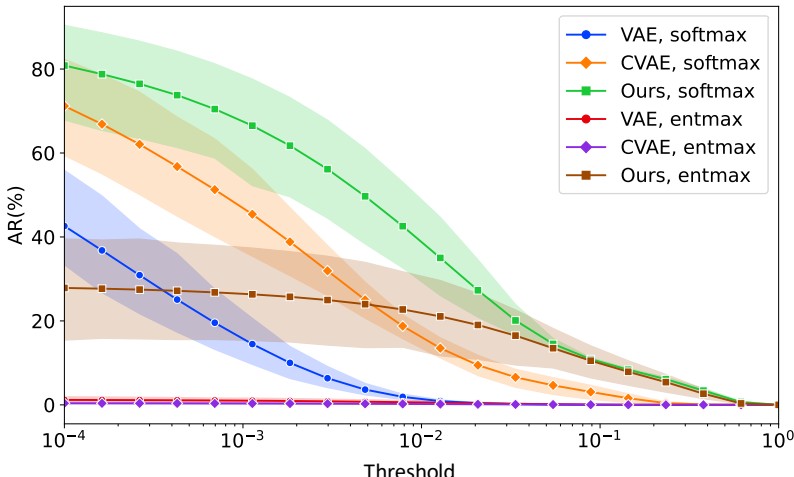

Figure 6: $\text{AR}_\delta(\%)$ vs. Threshold on the Argoverse Dataset.

Table 6: INTERACTION Dataset without Map - HM

| Map | Model | AR(%) | K = 1 | | K = 6 | |
| --- | --- | --- | --- | --- | --- | --- |
| | | | $min\text{FDE}$ | $min\text{ADE}$ | $min\text{FDE}$ | $min\text{ADE}$ |
| Yes | VAE | $8.68 \pm 8.36$ | $0.87 \pm 0.08$ | $0.29 \pm 0.03$ | $0.42 \pm 0.04$ | $0.16 \pm 0.01$ |
| | CVAE | $5.42 \pm 10.2$ | $0.83 \pm 0.03$ | $0.28 \pm 0.03$ | $0.40 \pm 0.05$ | $0.16 \pm 0.02$ |
| | Ours | $\mathbf{15.5 \pm 8.13}$ | $\mathbf{0.65 \pm 0.09}$ | $\mathbf{0.22 \pm 0.02}$ | $\mathbf{0.32 \pm 0.04}$ | $\mathbf{0.13 \pm 0.01}$ |
| No | VAE | $24.4 \pm 14.4$ | $0.98 \pm 0.08$ | $0.34 \pm 0.02$ | $0.51 \pm 0.05$ | $0.20 \pm 0.02$ |
| | CVAE | $\mathbf{43.5 \pm 8.72}$ | $0.72 \pm 0.02$ | $0.25 \pm 0.01$ | $0.36 \pm 0.01$ | $0.15 \pm 0.01$ |
| | Ours | $33.7 \pm 5.77$ | $\mathbf{0.62 \pm 0.01}$ | $\mathbf{0.22 \pm 0.01}$ | $\mathbf{0.32 \pm 0.01}$ | $\mathbf{0.13 \pm 0.01}$ |

information, which is verified by its lower AR value. In particular, one of the five VAE models gets nearly zero AR value.

The analysis becomes intricate when it comes to comparing our model against the CVAE variant. Introducing the auxiliary task still improves prediction accuracy. Also, we note that the CVAE model itself has a lower prediction error after removing lanelet nodes. If the historical social context is the dominating factor determining prediction performance, we may draw two conclusions. First, the CVAE model suffers less from social posterior collapse under the graph representation without lanelet nodes. Second, the auxiliary prediction task can further encourage the model to encode social context. However, we do not have direct evidence showing that the auxiliary task encourages the model to encode social context —the auxiliary task does not lead to a larger AR value.

We think the reason behind this is that removing lanelet nodes makes the attention map less likely to become sparse. In most driving scenarios, the number of lanelet nodes dominates the number of agents. Removing lanelet nodes reduces the number of incoming edges for each agent node. Consequently, the agent-to-agent edges compete with a single self-edge to gain attention instead of numerous lanelet-to-agent edges. It implies that the format of representation affects how the model suffers from social posterior collapse. More importantly, it shows the limitation of using AR as the single metric to analyze social posterior collapse. It is particularly effective when the agent vertices are of a high degree, but the analysis may become inconclusive otherwise. In future studies, we will investigate other metrics and tools that can be applied to a broader range of problems.

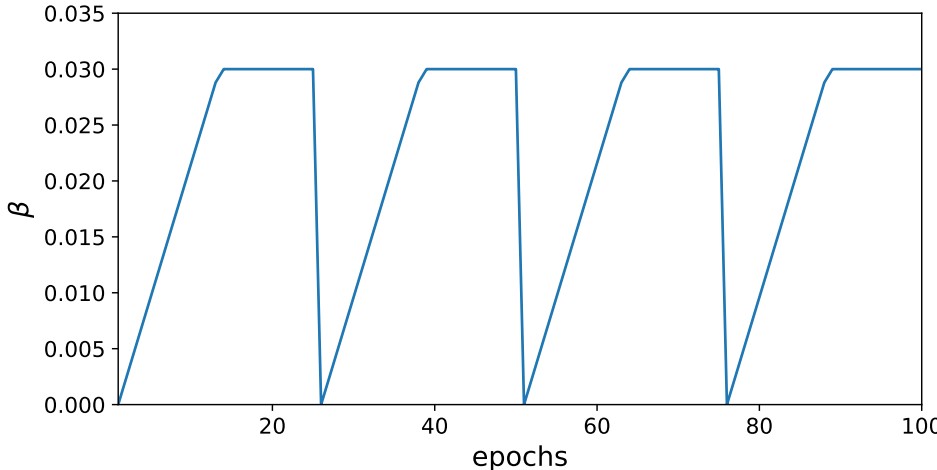

Figure 7: The cyclical annealing schedule adopted in our experiments. Each annealing cycle consists of twenty-five training epochs. The magnitude of $\beta$ increases linearly from zero to the maximum value in the first half of the cycle, and then keeps unchanged in the remaining epochs. The maximum value of is set to be $0.03$, which is the value used in previous experiments with constant $\beta$.

### E.3 Effect of KL Annealing

In Sec. 2.2, we argue that social posterior collapse is different from the well-known posterior collapse issue, and typical techniques that address posterior collapse, e.g., KL annealing, may not be effective in mitigating social posterior collapse. In this section, we test if one of the annealing methods —cyclical annealing schedule [19] —can effectively alleviate social posterior collapse. Again, we use the HM scenario from the INTERACTION dataset as an example to study this problem. When training the baseline VAE and CVAE models, we incorporate the cyclical annealing schedule plotted in Fig. 7 to adjust the magnitude of $\beta$ over training epochs. The results are summarized in Table 7. The cyclical annealing schedule neither improves prediction performance nor raises AR value consistently. The VAE model with a cyclical schedule does have a larger average AR value. However, two out of the five trials attains zero AR, which causes the large standard deviation. In summary, the cyclical annealing schedule does not alleviate the social posterior collapse issue in the experiments, which is consistent with our previous argument.

## F  Testing Results on Argoverse and ETH/UCY

In this section, we report the testing results on Argoverse and ETH/UCY datasets and compare the results with other models in the literature. It should be noted that achieving state-of-the-art performance is not our target. The testing results are provided to give the audience a complete picture

Table 7: INTERACTION Dataset KL Annealing Experiment - HM

| Model | Cycling | AR(%) | K = 1 | | K = 6 | |
|---|---|---|---|---|---|---|
| | | | $min$FDE | $min$ADE | $min$FDE | $min$ADE |
| VAE | No | $8.68 \pm 8.36$ | $0.87 \pm 0.08$ | $0.29 \pm 0.03$ | $0.42 \pm 0.04$ | $0.16 \pm 0.01$ |
| | Yes | $17.7 \pm 16.5$ | $0.96 \pm 0.10$ | $0.33 \pm 0.03$ | $0.52 \pm 0.04$ | $0.20 \pm 0.01$ |
| CVAE | No | $5.42 \pm 10.2$ | $0.83 \pm 0.03$ | $0.28 \pm 0.03$ | $0.40 \pm 0.05$ | $0.16 \pm 0.02$ |
| | Yes | $7.65 \pm 10.5$ | $0.84 \pm 0.07$ | $0.28 \pm 0.02$ | $0.41 \pm 0.03$ | $0.16 \pm 0.01$ |
| Ours | - | $\mathbf{15.5 \pm 8.13}$ | $\mathbf{0.65 \pm 0.09}$ | $\mathbf{0.22 \pm 0.02}$ | $\mathbf{0.32 \pm 0.04}$ | $\mathbf{0.13 \pm 0.01}$ |

Table 8: Argoverse Dataset Testing Results

| Model | K = 1 | | K = 6 | |
|---|---|---|---|---|
| | $min$FDE | $min$ADE | $min$FDE | $min$ADE |
| TNT [30] | 4.9593 | 2.1740 | 1.4457 | 0.9097 |
| LaneGCN [31] | 3.7786 | 1.7060 | 1.3640 | 0.8679 |
| LaneRCNN [32] | 3.6916 | 1.6852 | 1.4526 | 0.9038 |
| TPCN [48] | **3.6386** | **1.6376** | **1.3535** | **0.8546** |
| Social-CVAE (Ours) | 4.2748 | 1.9276 | 2.4881 | 1.3568 |

Table 9: ETH/UCY Testing Results

| Model | $min$ADE/$min$FDE, K = 20 | | | | | |
|---|---|---|---|---|---|---|
| | ETH | HOTEL | UNIV | ZARA1 | ZARA2 | Average |
| SGAN[2] | 0.81/1.52 | 0.72/1.61 | 0.60/1.26 | 0.34/0.69 | 0.42/0.84 | 0.58/1.18 |
| SoPhie[42] | 0.70/1.43 | 0.76/1.67 | 0.54/1.24 | 0.30/0.63 | 0.38/0.78 | 0.54/1.15 |
| Transformer-TF[49] | 0.61/1.12 | 0.18/0.30 | 0.35/0.65 | 0.22/0.38 | 0.17/0.32 | 0.31/0.55 |
| STAR[50] | 0.36/0.65 | 0.17/0.36 | 0.31/0.62 | 0.26/0.55 | 0.22/0.46 | 0.26/0.53 |
| PECNet[7] | 0.54/0.87 | 0.18/0.24 | 0.35/0.60 | 0.22/0.39 | 0.17/0.30 | 0.29/0.48 |
| Trajectron++[4] | 0.39/0.83 | 0.12/0.21 | **0.20/0.44** | **0.15**/0.33 | **0.11**/0.25 | 0.19/0.41 |
| AgentFormer[9] | **0.26/0.39** | **0.11/0.14** | 0.26/0.46 | **0.15/0.23** | 0.14/**0.24** | **0.18/0.29** |
| VAE (Ours) | 0.59/0.90 | 0.18/0.26 | 0.31/0.54 | 0.21/0.37 | 0.17/0.31 | 0.29/0.48 |
| CVAE (Ours) | 0.56/0.84 | 0.18/0.26 | 0.33/0.58 | 0.21/0.38 | 0.18/0.33 | 0.29/0.48 |
| Social-CVAE (Ours) | 0.64/0.99 | 0.18/0.27 | 0.35/0.62 | 0.21/0.37 | 0.19/0.34 | 0.32/0.52 |

of the model. For the Argoverse dataset, we collect the results of other models from the leaderboard. Even though there are other models on the leaderboard with better performance, we focus on those from published papers to get insights on the reasons behind the performance gap. For the ETH/UCY dataset, we collect the results from the corresponding papers.

On the Argoverse dataset, the performance of our social-CVAE model is similar to TNT [30], which adopted a similar map representation with us, in terms of $min$FDE and $min$ADE when K = 1. It implies that we could improve our performance by adopting alternative map representations, such as those proposed in LaneGCN [31] and TPCN [48]. Another observation is that compared to the other approaches, our model has a large performance margin in the case of K = 6. It is because sampling from a high-dimensional continuous distribution is inefficient, making it less effective in modeling the multi-modality of driving behavior. Using discrete latent space as in [6] and [4] or conditioning the latent space on goal points [7] could be better options under the CVAE framework. On the ETH/UCY dataset, the VAE variant of our model has better or comparable performance to all the other models except for Trajectron++ and AgentFormer. As mentioned in the main text, we are particularly interested in the formulation of AgentFormer. In future studies, we will incorporate a similar autoregressive decoder into our model and investigate social posterior collapse under this setting.