# OpenReview forum: "Exploring Social Posterior Collapse in Variational Autoencoder for Interaction Modeling"
_NeurIPS.cc/2021/Conference — NeurIPS 2021 Poster_

### Official Review · Reviewer_q4Va · 2021-07-09

**Rating:** 7
**Confidence:** 4

**Summary:**

This paper focuses on analyzing VAE-based frameworks for multi-agent trajectory forecasting from a relationship-modeling perspective. Specifically, it argues that standard VAE frameworks can potentially lead to a situation where the interactions between entities are ignored when predicting future trajectories, which they call “social posterior collapse.” To address this, they augment this framework in two ways: first, they convert the model to a conditional VAE (CVAE) where the prior on latent variables is conditioned on input trajectories; second, they add an auxiliary training task wherein samples from the prior are used to train a second trajectory decoder. To analyze the “social posterior collapse” issue, a graph attention layer that uses sparse attention is developed (titled “Sparse-GAMP”), and a metric called Agent Ratio (AR) is defined which consists of the proportion of surrounding agents to which the target agent is attending. Experiments are run on traffic and pedestrian trajectory prediction datasets, where it is shown that, in situations where social posterior collapse is shown to be present, their model is able to alleviate this issue. Furthermore, the auxiliary prediction task can lead to better model performance.

**Limitations And Societal Impact:**

The authors fully acknowledge that their results are not state-of-the-art, and they acknowledge that this is a limited study whose results might not apply to all models or domains - the results are not over-sold. There is no mention of potential negative societal impacts in this work.

**Main Review:**

### Originality:
the study undertaken in the paper is the first to my knowledge which attempts to explore the ability of VAE models for multiagent trajectory prediction tasks to properly model interactions between agents. The VAE framework that forms its basis is commonly used in prior work and appropriate for this study (and proper citations are made throughout).


### Quality:
Much care has been taken to develop the social-CVAE framework, from motivation through derivation. Additionally, the development of the AR metric to evaluate the ability of a model to model interactions between agents is useful for understanding the social posterior collapse issue. The experimentation is thorough, with multiple trials for every setting being run and important ablations being included in the appendix. Overall, the presented results are trustworthy, and the presented social-CVAE approach seems to mitigate social posterior collapse in situations where it appears in other approaches.

That being said, there is one set of experimental results presented in the supplementary material that ought to be mentioned in the main paper, as they alter the conclusions one should draw from this work. Specifically, the results presented in Table 6 indicate that the representation of context used by the model significantly impacts whether social posterior collapse occurs or not. When a lane graph-based representation is not included, no models suffer from social posterior collapse; furthermore, in this situation, the AR metric does not seem to correlate with the distance-based error metrics. These conclusions are additionally supported by the pedestrian dataset results presented in Table 3, where no map context is used. This dependence of social posterior collapse on context representation is important, but as far as I could tell was not mentioned in the main paper anywhere.



### Clarity:
The paper is written clearly, and is organized well. It is clear which components are drawn from prior work and which are novel developments. Sufficient detail is presented for reproducibility. As mentioned in the previous section, though, I think the results from Table 6 should be alluded to in the main paper.

### Significance:
As this paper cites, there are a number of papers which use VAE frameworks for multi-agent trajectory prediction. Hence, understanding the properties of these models, including how they model interactions between agents, is very important for the development of improved models. This paper takes a first step in this direction; though the results are not state-of-the-art, the analysis of social posterior collapse is interesting and may be useful for designing future models.


### Overall:
This work is a promising first step in understanding how to ensure interactions are encoded properly when using VAE-based multi agent trajectory prediction approaches. I do think that the lack of discussion regarding the impact of the context encoding representation hurts this paper somewhat, as it is possible to draw some incorrect conclusions without it.

#POST-REBUTTAL UPDATE
Thanks for your comments. Based on your responses, I am keeping my recommendation the same - I think this paper should be accepted. Please add the extra clarifications you mentioned in your response, as they will benefit the text.

**Time Spent Reviewing:**

2 hours

---

> ### Author Response · Authors · 2021-08-10
> **Author Response to Reviewer q4Va**
>
> Thank you so much for your comments. We are glad that you appreciate the novelty and value of our work.
> >“I do think that the lack of discussion regarding the impact of the context encoding representation hurts this paper somewhat, as it is possible to draw some incorrect conclusions without it.”
>
> The ablation study in E.2 is indeed a very important experiment. Thank you for pointing it out. We will summarize our findings of E.2 in the main text and draw our conclusion accordingly.
>
> >“When a lane graph-based representation is not included, no models suffer from social posterior collapse; furthermore, in this situation, the AR metric does not seem to correlate with the distance-based error metrics.”
>
> For the situation that AR metric does not correlate with the distance-based error metrics, we think the reason behind is that removing map context reduces the number of nodes in the graphs, which make the attention map less likely to become sparse. The proposed changes may still encourage the model to model social context, which results in the improvement in prediction metrics. In particular, only when all the changes are applied, the prediction result without map information is consistent with the case with map information. However, it is rather inconclusive as we cannot verify social posterior collapse through AR ratio. In Sec. E.2, we consider it as showing the limitation of our sparse graph attention in social posterior collapse detection. In our final version, we will summarize this limitation as well as other findings in E.2 in the main text.
>
> >“These conclusions are additionally supported by the pedestrian dataset results presented in Table 3, where no map context is used.”
>
> We would like to clarify that we think the case of pedestrians is different from the case of vehicles without map information. In the case of pedestrians, our social-CVAE model does not achieve better prediction performance. In the case of vehicles without map information, social-CVAE has the smallest prediction error, even if the AR ratio does not increase. Based on it and the nature of pedestrian interaction (i.e. historical social context does not affect long-term future motion), we think it is more likely due to the limitation of sparse-GAMP in social posterior detection. In the case of vehicles without maps, the model may still suffer from social posterior collapse, or at least, the historical social context is not properly modeled and utilized.

---

> > ### Comment · Reviewer_q4Va · 2021-08-27
> > **Keeping same recommendation**
> >
> > Thanks for the response. I've updated my review accordingly.

---

### Official Review · Reviewer_2YSC · 2021-07-17

**Rating:** 6
**Confidence:** 3

**Summary:**

The proposed paper introduces the concept of social posterior collapse in variational autoencoder (VAE) architectures trained for trajectory prediction in interactive environments. Social posterior collapse refers to the VAE ignoring interaction information during training, thus making trajectory predictions solely from the predicted agent's history and map information. The authors present a means to quantify the extent of social posterior collapse through a sparse attention mechanism and propose architectural changes to alleviate the issue.

**Limitations And Societal Impact:**

The authors did not indicate any negative societal impact to their work, but did contextualize the limitations of the work in great detail. I would encourage the authors to think about potential inadvertent consequences such as the amplification of implicit bias in the data in the process of sparsification or concerns regarding deployment of neural network architectures in safety critical systems.

**Main Review:**

I really like the new perspective taken by the authors in investigating existing difficulties of training VAEs in the specific context of trajectory prediction in interactive environments. The authors provide valuable insight with respect to the connections between general machine learning findings and trajectory prediction. The authors do an excellent job in contextualizing the work in terms of existing literature. I would recommend two additional papers in the space of sparse distributions and VAEs from NeurIPS last year to include in the literature review.

[1] Correia, Gonçalo, et al. "Efficient Marginalization of Discrete and Structured Latent Variables via Sparsity." Advances in Neural Information Processing Systems (2020).

[2] Itkina, Masha, et al. "Evidential Sparsification of Multimodal Latent Spaces in Conditional Variational Autoencoders." Advances in Neural Information Processing Systems (2020).

Employing sparse attention to quantify the extent of the social context used by the VAE to predict future trajectories is a good idea. The claim of quantifying the social context used to make predictions by the CVAE is well supported by the INTERACTION/Argoverse experiments. The authors also provide extensive experimental analysis in the supplementary material.

I appreciated the transparency and the insightful discussion dedicated to the limitations of the proposed method. Although portraying negative results is valuable, and a rare practice in machine learning papers, I am concerned that the pedestrian experiments actually detract from and confuse the message of the paper in the current framing. On page 2 of the paper, the authors frame the task of interest as predicting multiple trajectories simultaneously (lines 40-41). The problem with current trajectory prediction models is defined as their tendency to ignore interactions over the predicted horizon (lines 72-75). However, the successful experiments focus on a single vehicle trajectory prediction task, demonstrating that the interaction history is important for the prediction. The unsuccessful pedestrian dataset experiments, despite providing significant insight into both the dataset and the problem, do not validate the original hypothesis of the paper. I would reframe the beginning of the paper to focus on ensuring that the \textit{historical social context}, rather than future interaction, is accounted for in the models for predicting single trajectories at a time. That way the successful experiments validate this idea. The pedestrian experiments can be kept as a case study extension in the main paper. Alternatively, successful experiments demonstrating the simultaneous prediction of all trajectories in the scene should be included in the paper, and the pedestrian experiments moved to supplementary.

I have some concerns with respect to the clarity of some of the presented material. Could you clarify the connection between the agent nodes and the $T_{i}$ context encodings? Since $T_{i}$ incorporates information from all of the observed agents by agent $i$, while sparse attention determines which agent connections are being used by the model, I would like to better understand the relationship between the two concepts.

Several variables were not defined (e.g., $\mathbf{s}$ on line 158, $I$ on line 110). It is great to see that the authors did their due diligence by running the experiments on multiple random seems. However, I have some questions regarding the standard deviation reported in Tables 1-3, but particularly Table 1. First, are these standard deviations or standard error? The latter is usually reported as a metric of result significance. Second, if these are standard errors, they are rather large, potentially demonstrating the results to not be significant? Third, why are there standard errors reported for only one of the columns in Table 1? Also in Table 1, could you clarify what the difference was between the models 'Ours' and 'Ours*'? I am not sure what the significance of the discussion of the latter model is in lines 261-265?

The presentation of the paper needs improvement in that it is riddled with typos and grammatical errors. Some examples are as follows, but these are far from exhaustive:

- 'the model is prone to ignore social context' (should be 'prone to ignoring').
- 'a model that models each agent's behavior separately without social context. which might suffer from over-estimated variance and large prediction error.' (punctuation/half-sentence typo).
- 'to interfere the decoding procedure' (missing 'with').

The references should also be proofread for consistency in conference naming, capitalization (e.g., 'gan' -> 'GAN', 'bayes' -> 'Bayes'), etc.

Overall, the authors present an interesting approach to evaluating social posterior collapse in VAE models for trajectory prediction. I really liked the idea and the in-depth discussion provided in the paper. With improvements to the presentation of the paper and the framing of the experiments with respect to the problem statement, this work will be a good contribution!

**Time Spent Reviewing:**

8

---

> ### Author Response · Authors · 2021-08-10
> **Author Response to Reviewer 2YSC**
>
> Thank you very much for your detailed review and valuable suggestions. We are encouraged that you like the new perspective we took in this study and find the sights given in the paper valuable.
> >“I would recommend two additional papers in the space of sparse distributions and VAEs from NeurIPS last year to include in the literature review.”
>
> Thank you for your suggestions. We will review them in our final version when introducing sparse attention functions.
>
> >“I would reframe the beginning of the paper to focus on ensuring that the $\textit{historical social context}$, rather than future interaction, is accounted for in the models for predicting single trajectories at a time. That way the successful experiments validate this idea. The pedestrian experiments can be kept as a case study extension in the main paper.”
>
> Thank you for your valuable suggestion. For clarification, we mention in the first paragraph of Sec. 2.1 that the VAE framework studied in this paper does not explicitly model future interaction. We think you make a valuable point that we should reframe the beginning of the paper to focus on the settings corresponding to vehicle prediction tasks. In response to this, we will explicitly confine the social posterior collapse phenomenon studied in this paper as ignoring historical social context. We will state in Introduction and Sec. 2.2 that if the agents in an interacting system are affected by their historical social context, then social posterior collapse may damage model performance. In this case, the pedestrian scenario is considered as a case where the assumption (i.e. historical social context matters) might not be satisfied according to the experimental results and literature. The pedestrian experiment will be treated as a case study when the condition is not met. Also, it will be used to introduce a future extension to the current study: how to ensure future interaction is properly modeled for those interacting systems where future social context is important.
>
> >“Could you clarify the connection between the agent nodes and the Ti context encodings? Since Ti incorporates information from all of the observed agents by agent i, while sparse attention determines which agent connections are being used by the model, I would like to better understand the relationship between the two concepts.”
>
> In the input graphs, each agent node has the agent’s state $x_i$ as the node feature. In the sparse-GAMP layer, the sparse attention determines which agent connections are used and aggregates those activated agents’ states into the context embeddings $T_i$. After the sparse-GAMP layer, each agent node will have $T_i$ as the node feature.
>
> >“First, are these standard deviations or standard error? The latter is usually reported as a metric of result significance. Second, if these are standard errors, they are rather large, potentially demonstrating the results to not be significant?Third, why are there standard errors reported for only one of the columns in Table 1?”
>
> Those are standard deviations that are reported in Tables 1-3. We mention it in line 251. In the second half of Table 1, the metrics reported are results on the testing set. We get these results by uploading our predictions to the official challenge held for INTERACTION. To follow a practice to avoid using testing sets for validation, we just select one single model according to validation for submission. That’s why we don’t report the standard deviation. The standard deviation of AR is computed over all the testing samples, so we still have a standard deviation for AR. We are sorry that we do not explain well how we compute the standard deviation of AR. We will make sure it is clear in the final version.
>
> >“Also in Table 1, could you clarify what the difference was between the models 'Ours' and 'Ours*'? I am not sure what the significance of the discussion of the latter model is in lines 261-265?”
>
> We present this result to address the potential concern that the proposed changes in model design can improve performance, but they do not improve performance through mitigating social posterior collapse. The model ‘Ours*’ is an instance where the AR value is close to zero (i.e. social posterior collapse occurs) even with all the changes. Its performance is worse than the ‘Ours’, an instance with large AR value. We can then identify mitigating social posterior collapse as the main cause of performance improvement.
>
> For all the remaining typos, errors, and missing definitions, we will correct them in our final version. Thank you for pointing them out.

---

> > ### Comment · Reviewer_2YSC · 2021-08-30
> > **Keeping the original score**
> >
> > Thank you to the authors for the clarifications! The paper would benefit from including these clarifications and structure changes in the final version. Overall, I will keep my score in leaning towards accept.

---

### Official Review · Reviewer_8WnM · 2021-07-17

**Rating:** 4
**Confidence:** 4

**Summary:**

This paper analyzes the reason behind the social posterior collapse caused in typical VAEs for multi-agent modeling and proposes improvement schemes. The paper test the proposed model on real-world multi-agent trajectory prediction datasets. This paper also proposes a novel sparse graph attention message-passing.

**Limitations And Societal Impact:**

1 What's the meaning of Position 2 in the sparse graph attention message-passing towards social posterior collapse detection?

2 The inferred sparse attention is only used in e2v process. The v2e process still causes large storage and complexity.

3 What is the effects of the sparse graph attention message-passing, compared to the ordinary neural-message-passing without sparse attention? From the aspects of error and efficiency?

4 Many excellent previous works are not introduced for performance comparison, some of which are based on CVAE or neural-message-passing, including:

[1] Social lstm:  Human trajectory prediction in crowded spaces

[2] Social attention: Modeling attention in human crowds

[3] Social gan:  Socially acceptable trajectories with generative adversarial networks.

[4] Stgat: Modeling spatial-temporal interactions for human trajectory prediction

[5] Sophie: An attentive gan for predicting paths compliant to social and physical constraints

[6] Multi-agent tensor fusion for contextual trajectory prediction

[7] Encoding crowd interaction with deep neural network for pedestrian trajectory prediction

[8] Collaborative motion prediction via neural motion message passing

[9] Trajectron++:  Multi-agent generative trajectory forecasting with heterogeneous data for control

[10] Conditional flow variational autoencoders for structured sequence prediction.

[11] Neural relational inference for interacting systems.

[12] Social-stgcnn: A social414spatio-temporal graph convolutional neural network for human trajectory prediction.

The results in this paper still need to be improved

**Main Review:**

originality: good
quality: need to improve
clarity: some designs are not easy to understand
significance: just ok


**Time Spent Reviewing:**

1 day

---

> ### Author Response · Authors · 2021-08-10
> **Author Response to Reviewer 8WnM**
>
> Thank you so much for your feedback. Here are our thoughts on your four comments:
>
> >“What's the meaning of Position 2 in the sparse graph attention message-passing towards social posterior collapse detection?”
>
> Proposition 2 mainly shows two properties of 1.5-entmax. The first statement shows a sufficient condition for augmenting an input vector without affecting its original attention values. We use it to derive an efficient and convenient implementation of the sparse graph attention function, where it needs to handle inputs with varying dimensions in one batch. The trick makes it more efficient to use the sparse-GAMP layer in our experiments. The second statement shows that the activated coordinates determine a unique threshold value for augmented coordinates. That is to say, if we apply it to model the interaction between an agent with other surrounding agents, then the attention weights assigned to those agents that truly interact with the modeled agent will not be diluted by the irrelevant agents. We think it could potentially improve the robustness of the model especially when the number of agents is large (e.g., dense traffic scenes). It is an interesting insight we found when exploring the sparse graph attention function. We do not utilize this property in this work as it is not related to the study of social posterior collapse, but we are interested in investigating its application in future work.
>
> >“The inferred sparse attention is only used in e2v process. The v2e process still causes large storage and complexity.”
>
> Yes, the sparse attention is only applied in the e2v process. Compared to ordinary graph attention, our sparse attention does not have advantages in terms of storage and complexity. However, it is not the motivation for us to use sparse-GAMP over other GNNs, and we never claim it in our paper. We think the sparse graph attention is better than other aggregation functions in social posterior collapse detection, because it induces a sparse and interpretable attention map. It enables us to use the percentage of unattended agent-to-agent edges as a metric to detect social posterior collapse, which is the Agent Ratio (AR) metric introduced in Sec. 5.1.
>
> >“What is the effects of the sparse graph attention message-passing, compared to the ordinary neural-message-passing without sparse attention? From the aspects of error and efficiency?”
>
> As we said, we use sparse-GAMP over other message-passing layers mainly for the sparse and interpretable attention map it induces. We do not claim it has advantages in terms of performance and efficiency. For your reference, we do include an ablation study in the supplementary material (Sec. E.1), where we replace the sparse graph attention with other aggregation functions. However, the focus of that ablation study is really to verify that sparse-GAMP itself is not the reason behind social posterior collapse and it does not interfere with prediction performance, so that we can safely use it as a tool to detect social posterior collapse.
>
> >“Many excellent previous works are not introduced for performance comparison, some of which are based on CVAE or neural-message-passing.”
>
> Thank you for bringing all these excellent prior works to our attention. We would like to clarify that our focus is to study social posterior collapse in the VAE framework introduced in Sec. 2.1 for interaction modeling. The main objective of our experiments is to study whether social posterior collapse can be mitigated by the two proposed measures: 1) changing the model framework from VAE to CVAE; 2) adding an auxiliary prediction task. These measures are described in the last three paragraphs of Section 2.2, starting from line 123. The sparse graph attention message-passing layer is proposed as a tool to detect and analyze social posterior collapse. We do not claim that sparse-GAMP itself can mitigate social posterior collapse or improve model performance. Therefore, our experiments mainly focus on comparing model variants with and without those proposed measures. Besides, we state explicitly in Sec. 6 that the claims and analysis in this paper are limited to the specific formulation we study.
>
> Among the listed papers, [1 - 8] and [12] do not use VAE. Since our focus is to study social posterior collapse in VAE models for interaction modeling, comparing the model performance with those non-VAE models does not help validating our claims. [9-11] indeed relate to both VAE and interaction modeling. However, as the first study on this topic, we would like to have an in-depth exploration under one setting. Studying social posterior collapse in a different framework needs deliberate analysis and experiment design, which could be too overwhelming for a single paper.
>
> We do include a review of prior works on VAE-based interaction modeling in Sec. 6. We focus on those works with elements that may implicitly tackle social posterior collapse. We would like to explicitly study social posterior collapse under their settings in the future. [9] is included in our discussion, as we think the discrete latent space they use may mitigate social posterior collapse. [11] also adopts a discrete latent space. We do cite [11] somewhere else in the paper. In the final version, we will include [11] in Sec. 6 when mentioning the potential effect of discrete latent space. For [10], the main focus of theirs is to address the posterior collapse problem in order to capture multi-modality. It is different from the social posterior collapse phenomenon we define and study in this paper. It is also not clear to us if the proposed regularization schemes in their work may mitigate social posterior collapse.
>
> As a side note, we include a comparison on ETH/UCY dataset (Table 9 in Appendix) between our models with other state-of-the-art methods, including [3], [5] and [9]. However, it is merely to give the audience a complete picture of the model. It is not related to our claims in the main text.

---

### Official Review · Reviewer_3CCN · 2021-07-18

**Rating:** 6
**Confidence:** 2

**Summary:**

This work discuss the problem of posterior collapse in VAE/CVAE problems and extend this when it is used in the concept of multi-agent  interaction modelling and trajectory prediction problem. They argues that the model designed naively based on VAE for this task is prone to ignore agent interaction when predicting the future trajectory of an agent. The framework proposes a novel sparse graph attention message-passing (sparse-GAMP) layer to address this problem.

**Limitations And Societal Impact:**

No, they haven't.

**Main Review:**

The problem discussed in this paper is very interesting and the authors well justified this critical issue with VAEs theoretically. My main concern is that the experiments and evaluations are not sufficient to validate the problem and efficacy of the solution. I suggest that the paper should include one experiment without any joint (social) interaction (encoding and decoding for each agent independently) and proves that the results using VAE in social setting is the same (or even worse) than this baseline. Also to show the efficacy of the solution, the authors could try their solution in many frameworks using VAE (trajectron, trajectron++ etc) and show that their sparse-GAMP layer can improve the performance of the state-of-the-art methods for this problem. Without these experiments, it is hard to support the acceptance of the paper with the theoretical arguments and a weak experimental validation only.

**Time Spent Reviewing:**

2

---

> ### Author Response · Authors · 2021-08-10
> **Author Response to Reviewer 3CCN**
>
> Thank you so much for your comments and suggestions on our paper. We are glad that you found this paper interesting and appreciate our effort in justifying this critical issue.
>
> >“The framework proposes a novel sparse graph attention message-passing (sparse-GAMP) layer to address this problem.”
>
> We would like to clarify that we propose to mitigate social posterior collapse by two approaches : 1) changing the model framework from VAE to CVAE; 2) adding an auxiliary prediction task. These measures are described in the last three paragraphs of Section 2.2, starting from line 123. The sparse graph attention message-passing layer is proposed as a tool to detect social posterior collapse. It is used in our experiments to analyze if the proposed approaches can indeed mitigate social posterior collapse. We do not claim that sparse-GAMP itself can mitigate social posterior collapse or improve model performance.
>
> >“I suggest that the paper should include one experiment without any joint (social) interaction (encoding and decoding for each agent ??>Independently) and proves that the results using VAE in social setting is the same (or even worse) than this baseline.”
>
> Could you clarify how the suggested baseline without social interaction can help validate the problem? We are particularly confused why the VAE with social interaction is expected to perform even worse than the baseline without social interaction.
>
> >“Also to show the efficacy of the solution, the authors could try their solution in many frameworks using VAE (trajectron, trajectron++ etc) ?>and show that their sparse-GAMP layer can improve the performance of the state-of-the-art methods for this problem.”
>
> We are indeed interested in exploring social posterior collapse in other frameworks using VAE in our future work (but we are not meant to show sparse-GAMP can improve the performance of those state-of-the-art methods, as we said before). We actually mention a few of them (including trajectron, trajectron++) in Section 6 and explain why we are interested in studying social posterior collapse in their settings. However, our focus of this paper is to study social posterior collapse under the VAE framework formulated in Sec. 2.1. We deliberately emphasize this point in Sec. 6. As the first study on this topic, we would like to have an in-depth exploration under one single setting. Studying the social posterior collapse problem under a different framework needs deliberate analysis and experiment design. For instance, as we mention in Sec. 6, Trajectron and Trajetron++ might tackle this issue implicitly, because they use a discrete latent space instead of a continuous one. We would still like to study explicitly if social posterior collapse occurs and if the discrete latent space can mitigate social posterior collapse. That will involve a different set of theoretical tools and experimental designs, which could be too overwhelming to be included in this paper.

---

> > ### Comment · Reviewer_3CCN · 2021-09-01
> > **Keep my rating**
> >
> > Thanks to the authors for their clarifications about the first and second comments. However, I still believe that in order to validate their hypothesis, the authors could demonstrate this problem in many more frameworks using VAE in social interaction context (trajectory prediction, human motion interaction etc). This is an interesting paper, but experiments could be improved. To this end, I keep my current rating

---

### Decision · Program_Chairs · 2021-09-27

**Decision:**

Accept (Poster)

**Comment:**

The paper identifies a failure mode of a common approach to modeling interaction using Variational Autoencoders in multi-agent behavior and trajectory forecasting, which the authors term "Social Posterior Collapse". This identified tendency to underfit in settings where relevant social context could improve prediction and generalization performance. The authors propose an approach to solving this problem using a contextual VAE formulation and auxiliary prediction task, and demonstrate that these lead to the expected empirical improvements in multi-agent interaction settings.


Reviewers assessed the focus of the papers as an important and interesting problem that is well diagnosed theoretically. Given that VAEs are commonly used to model multi-agent interactions, the identified failure mode, proposed metric for diagnosing it, and the novel solution to this problem have good potential for impact.

A number of concerns were raised in the initial reviews. In particular, reviewers suggested additional empirical validation to more clearly pin point the failure case, and to probe its generality as well as the generality of the new proposed solution. In addition, reviewers made valuable suggestions for improving clarity as well as further increasing the contribution of the paper by moving an experiment from the appendix to the main paper.

In the rebuttal and discussion, reviewers indicated that their concerns were largely addressed. The most negative reviewer unfortunately did not follow up on the discussion. The AC reviewed their concerns and the author response, and assessed the concerns as sufficiently addressed. While the paper can be improved further, remaining issues can be addressed in the camera ready version and do not prevent acceptance of the paper. The AC recommends that the paper is accepted, and strongly encourages the authors to take on board all reviewer suggestions to further improve the camera ready version.